# Nicotine biosynthesis is completed by cryptic activating glucosylation

Benjamin T. W. Schwabe [1,2], Isabelle M. Angstman[3], Katharina Vollheyde [3], Zoe Ingold [1,2], Jiacheng Li[2], Ksenia S. Stankevich [2,4], Christopher D. Spicer [2,4], Martin A. Fascione [2,4], Gideon Grogan [2], Fernando Geu-Flores [3] & Benjamin R. Lichman [1] ✉

Nicotine is a neuroactive alkaloid produced by tobacco (*Nicotiana tabacum*) as a defence against herbivory, and an addictive stimulant that has been used by humans for millennia. Despite its significance, the core steps of its biosynthesis have remained elusive. Here, we demonstrate the in vitro reconstruction of a four-enzyme stereoselective biocatalytic cascade that forms (*S*)-nicotine from nicotinic acid and *N*-methylpyrrolinium. This cascade includes two glucose-processing enzymes that participate in a cryptic activating glucosylation step. We also reconstruct this pathway *in planta* and present high resolution X-ray structures of the key carbon-carbon bond forming reductase-oxidase pair bound to their substrate and product, respectively. This work establishes the complete biosynthetic pathway to nicotine, providing new gene targets for controlling alkaloid production in *Nicotiana* and unlocking enzymatic routes to pyridine alkaloids.

Nicotine (**1**) is a plant-derived alkaloid of profound societal and scientific importance. It is the major neuroactive and addictive component of cultivated tobacco (*Nicotiana tabacum*) products, which have been used by humans for over 10,000 years[1] and are responsible for an ongoing global health epidemic[2]. Its neurological properties are due to its agonistic interaction with eponymous nicotinic acetylcholine receptors[3]. The alkaloid protects plants from herbivory[4] and its insecticidal activity led to its use as an agricultural pesticide, prior to replacement by neo-nicotinoids[5].

Tobacco and its alkaloids have formed the basis of extensive investigations into plant biology including chemical ecology[6], metabolite transport[7], genetic regulation[8] and metabolic evolution[9]. Recently, tobacco species, especially *Nicotiana benthamiana*, have gained traction as chassis for the recombinant production of therapeutic molecules and proteins, including vaccines[10]. However, the presence of nicotine can cause challenges for downstream processing[11]. Despite its notability, nicotine's biosynthesis has not been elucidated: solving it will enable genetic manipulation of the pathway in native and heterologous producers, and will provide biocatalytic tools for the formation of pyridine-containing compounds.

Alkaloid scaffolds are typically formed via Mannich-like reactions, wherein an iminium electrophile is attacked by a carbon nucleophile, forming a carbon-carbon bond and a chiral center[12]. Nicotine (**1**) consists of pyridine and *N*-methylpyrrolidine rings, derived from nicotinic acid (**2**) and *N*-methylpyrrolinium (**3**) respectively (Fig. 1)[13,14]. Labelled precursor feeding studies indicate that **2** undergoes C6-reduction and decarboxylation to yield the unusual nucleophile 1,2-dihydropyridine (**4a**), which reacts at C3 with the electrophilic iminium **3** to form 3,6-dihydronicotine (**5a**) followed by stereoselective removal of the C6-hydrogen to form **1** (Fig. 1, red hydrogen)[15–22]. The formation of **1** appears to be stereoselective, as the (*S*)−**1** enantiomer makes up 96% of **1** formed in *N. tabacum* roots, prior to its enrichment in leaves through enantioselective demethylation of (*R*)−**1**[23]. Related pyridine alkaloids anabasine (**6**) and anatabine (**7**) are formed when **3** is replaced by lysine-derived Δ¹-piperideinium (**8**) or nicotinic acid-derived 2,5-dihydropyridinium (**9**) respectively (Fig. 1)[24].

---

[1]Centre for Novel Agricultural Products, Department of Biology, University of York, York, UK. [2]Department of Chemistry, University of York, York, UK. [3]Section for Plant Biochemistry and Copenhagen Plant Science Centre, Department of Plant and Environmental Sciences, University of Copenhagen, Frederiksberg, Denmark. [4]York Biomedical Research Institute, University of York, York, UK. ✉e-mail: benjamin.lichman@york.ac.uk

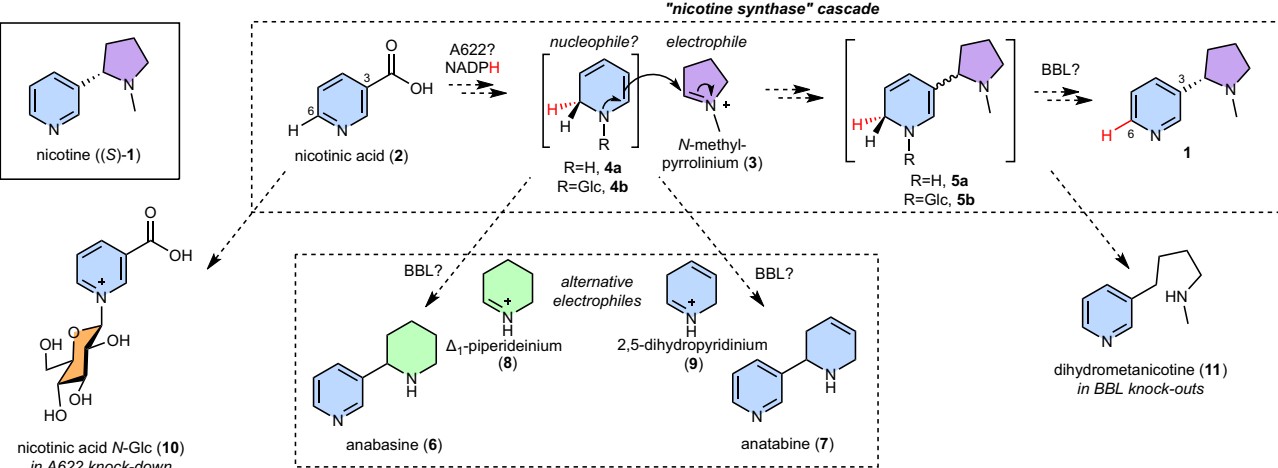

**Fig. 1 | Proposed biosynthesis of nicotine.** Hypothesized biosynthetic route to (S)-nicotine ((S)−**1**), showing origins of the pyridine (blue) and N-methylpyrrolidine (purple) rings. The Mannich-like scaffold formation is proposed to proceed via a multi-step 'nicotine synthase' cascade[16]. Compounds that accumulate in *Nicotiana* silencing/knock-out experiments are depicted. The red hydrogens follow the fate of the hydride transferred via enzymatic reduction of **2**.

In 1990, the isolation of 'nicotine synthase' was reported, a crude enzyme preparation from *N. tabacum* capable of forming (S)−**1** from **2** and **3**[16]. Subsequently, A622 and BBL were described, two oxidoreductases involved in nicotine scaffold formation, first identified in low-nicotine tobacco mutants[25–31]. However, their substrates and products have not been determined. Silencing of *A622* and its close homolog *A622L* in hairy root cultures led to a reduction in pyridine alkaloids and an increase in nicotinic acid N-glucoside (**10**) (Fig. 1)[26], suggesting that A622, a cytosolically localised isoflavone reductase-like enzyme[32], is responsible for the reduction of **2** or a derivative. However, recombinant enzyme assays failed to demonstrate activity of A622 with **2** or **10**[26]. *BBLs* encode vacuolar-localised flavin-containing oxidases from the berberine bridge enzyme family[33]. RNAi-based suppression or CRISPR-based inactivation of *BBLs* led to a decrease in alkaloid content and an accumulation of dihydrometanicotine (**11**), which may be derived from dihydronicotine (**5a**) (Fig. 1)[30,31]. Furthermore, residual **1** in quintuple *BBL* knockouts was racemic and displayed a labelling pattern consistent with disruption of stereoselective H-loss at C6[31]. Single gene knockouts suggest that *BBLa* and *BBLb* are responsible for (S)−**1** formation whereas *BBLc* may be responsible for (R)−**1** formation[34]. Whilst it appears that BBL paralogs have roles in both oxidation and stereochemical control, the accumulation of **11** in knockouts suggests they are not responsible for the ring coupling step.

It is nearing two hundred years since the first isolation of nicotine, in 1828, by Posselt and Reimann[35]. Since then, the nicotine biosynthesis pathway has been under intense scrutiny due to its commercial and scientific significance. Nevertheless, progress in defining the crucial step in its formation—the coupling of the two rings—has been remarkably static, with current conceptions remaining essentially unchanged since those proposed in the 1960s[15].

In this work, we resolve this enduring mystery, discovering two glucosyl-processing enzymes that complete nicotine biosynthesis via a cryptic activating glucosylation. We demonstrate this by reconstructing in vitro a four-enzyme stereoselective biocatalytic cascade that forms (S)-nicotine from nicotinic acid and N-methylpyrrolinium. Furthermore, X-ray crystal structures of the key oxidoreductases, alongside *in planta* pathway reconstruction and metabolite analysis, reveal the molecular and enzymatic basis for nicotine biosynthesis.

## Results
### Proposal of glucosylated intermediates in nicotine biosynthesis
Genes involved in plant specialised metabolism pathways can be arranged in biosynthetic gene clusters, where non-homologous but functionally related genes are found in close genomic proximity[36]. We investigated the genomic context of nicotine biosynthesis genes (Supplementary Data 1), and identified a cluster consisting of *A622* (*884g0010* from Edwards et al.[37]), the nicotine-related tonoplastic *MATE1* transporter (*884g0030*)[26,38,39] and a previously unreported gene predicted to encode a vacuolar localised β-glucosidase (*β-GD1*, *884g0020*) (Fig. 2A and Supplementary Fig. 1). We identified the homeologous cluster on chromosome 12, featuring *A622L* (*31g0370*), *MATE2* (*31g0420*) and a β-GD paralog (*β-GD2, 31g0400*) (Fig. 2A). All clustered genes, as well as an unclustered β-GD paralog (*β-GD3, 795g0060*) (Supplementary Data 1), had highest expression in roots and co-expressed tightly with known nicotine biosynthesis genes (Fig. 2B, C and Supplementary Data 2)[40].

The clustering and co-expression pattern of the β-GDs suggested they are involved in nicotine biosynthesis, implying on-pathway deglucosylation. This led us to reconsider nicotinic acid N-glucoside (**10**) as the substrate of A622, hypothesising that 1,2-dihydropyridine glucoside (**4b**, R=Glc) and 1,2-dihydronicotine glucoside (**5b**, R=Glc) may be pathway intermediates (Fig. 1). Therefore, the remaining step requiring a candidate gene was the glucosylation of nicotinic acid (**2**). We identified a UDP-dependent glycosyltransferase (UGT) candidate (*UGT1, 6222g0020*) whose expression mirrored *A622* expression after topping, a process that elicits nicotine biosynthesis (Fig. 2D)[41]. *UGT1* expression closely correlates with nicotine biosynthesis genes across multiple tissues (Fig. 2B, C and Supplementary Data 3). It was given the systematic UDP-glycosyltransferase name *UGT709L18*. Having identified β-GD and UGT candidates and proposed functions for A622 and BBL, we set out to test our hypothesis through pathway reconstruction.

### In vitro reconstitution of nicotine biosynthesis
We first reconstructed the nicotine biosynthesis pathway in vitro, selecting this simple system as it lacks interfering enzymes, non-relevant metabolites and complications of subcellular and tissue localisation. For expediency, we elected to test a single representative paralog of each enzyme type. Four heterologously expressed and purified enzymes (UGT1, A622, BBLa and β-GD1, Supplementary Data 4, and Supplementary Fig. 2) were combined in one-pot reactions alongside co-substrates (UDP-glucose, NADPH) and precursors (Fig. 3). In the presence of all four enzymes, (S)-nicotine ((S)−**1**) was formed from nicotinic acid (**2**) and N-methylpyrrolinium (**3**), reconstituting the activity previously observed in the 'nicotine synthase' crude enzyme preparation (Figs. 3; 4A and Supplementary Fig. 3)[16].

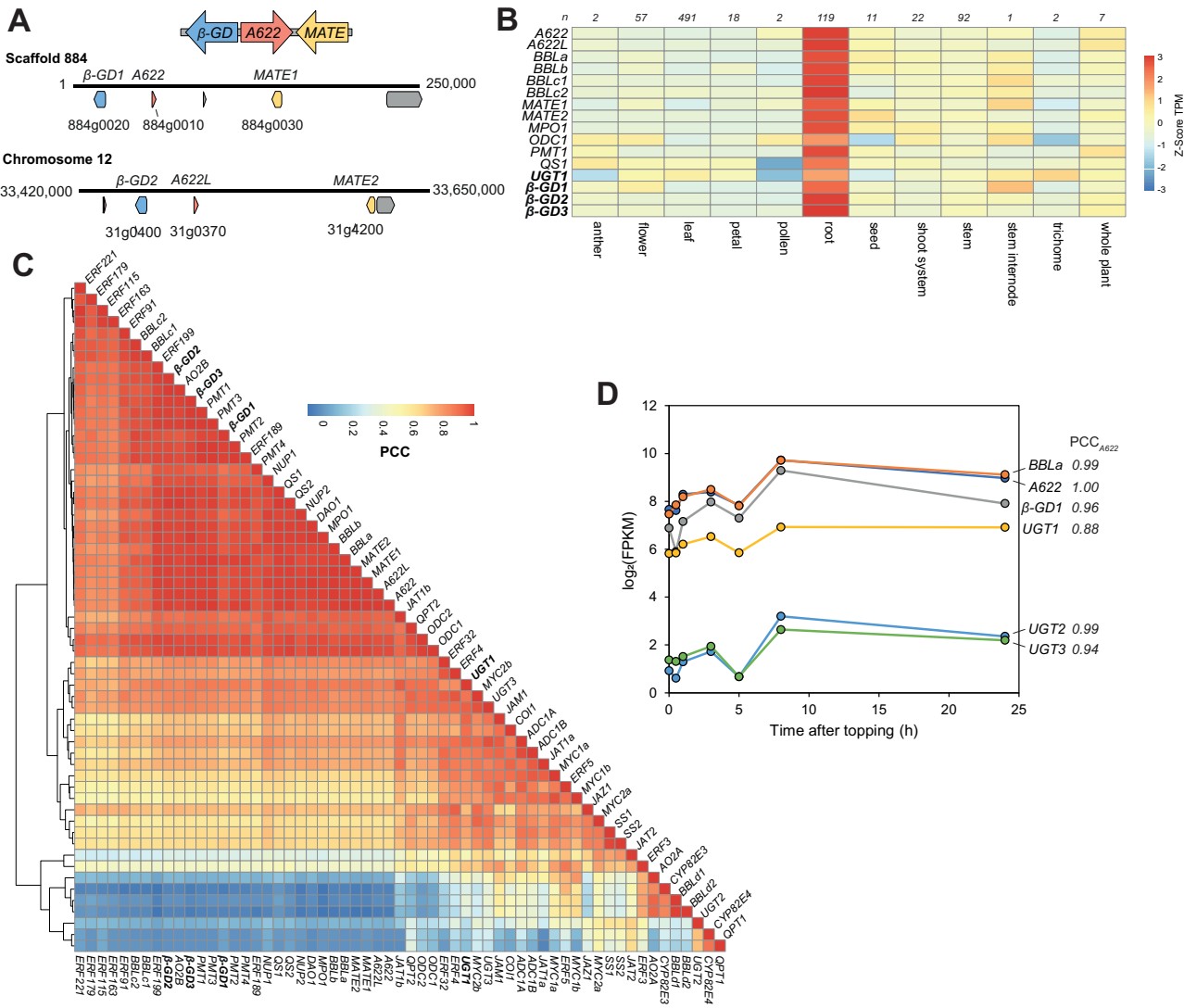

**Fig. 2 | Identification of biosynthetic gene clusters and gene candidates.**
**A** Homeologous biosynthetic gene clusters identified in the *N. tabacum* 2017 genome assembly[37] consisting of β-glucosidase (*β-GD*, blue), reductase (*A622*, red) and transporter (*MATE*, yellow) genes. Scaffold/chromosome number is described above the linear genome strand, with numbers at either end representing nucleotide position. Scaffold 884 is unplaced in this assembly but in the Sierro et al. 2024 assembly it is part of chromosome 16[75]. The gene model between *A622* and *MATE1* in scaffold 884 (*884g0040*) encodes a protein that contains a domain with homology to a reverse transcriptase domain (PANTHER ID PTHR33116). **B** Heatmap of expression of selected nicotine genes, including *β-GD* and *UGT* candidates across tissue types. Average expression calculated in each tissue type and values normalized per gene. Number of samples in each category above each column. **C** Clustered heatmap showing all-by-all correlations (PCC) of nicotine biosynthesis and related genes across 824 samples. TPM values obtained from PEO[40]. **D** Root expression levels of selected nicotine biosynthesis genes and UGT candidates following topping treatment, and their correlation (PCC) with A622 responses. Data obtained from Qin et al. 2020[41]. Details of genes available in Supplementary Data 1.

The UGT1 enzyme was determined to be a nicotinic acid *N*-glucosyltransferase (NaGT), producing nicotinic acid *N*-glucoside (**10**) from nicotinic acid (**2**) and UDP-glucose (Fig. 3 and Supplementary Fig. 4). NaGT could also catalyse the formation of pyridine glucoside (**12**) from pyridine and UDP-glucose, but could not form nicotine glucoside (**13**) from nicotine and UDP-glucose (Supplementary Fig. 4).

A622 was determined to be a nicotinic acid *N*-glucoside reductase (NaGR), catalysing the NADPH-dependent reduction of **10** (Fig. 3 and Supplementary Fig. 5), with the inferred product 1,2-dihydropyridine glucoside (**4b**) undergoing non-enzymatic oxidation to yield pyridine glucoside (**12**). No peak corresponding to the expected mass of 1,2-dihydropyridine glucoside (**4b**) ([M + H]⁺ = *m/z* 244) could be observed in NaGR-catalysed reactions. The formation of **4b** from dihydronicotinic acid *N*-glucoside (**14**) could occur non-enzymatically through an enamine-imine tautomerisation followed by

decarboxylation (Supplementary Fig. 5C). The subsequent oxidation of **4b** to form **12** likely occurs through reaction with molecular oxygen.

When *N*-methylpyrrolinium (**3**) was included in reactions with NaGR and **10**, the levels of **12** decreased, and a diastereomeric mixture of (*R, S*)-nicotine glucoside ((*R, S*)−**13**) appeared (Figs. 3 and 4B). This is likely due to a non-stereoselective Mannich reaction between **3** and **4b**, forming (*R, S*)-dihydronicotine glucoside (**5b**), which undergoes non-enzymatic oxidation to yield (*R, S*)−**13**. We were also able to identify a small peak matching the expected mass of dihydronicotine glucoside (**5b**) ([M + H]⁺ = *m/z* 327) but this peak was assigned as its isomer dihydrometanicotine *N*-glucoside (Glc-**11**) via comparison to a chemically synthesised standard (Supplementary Fig. 5). We also identified traces of the aglycone dihydrometanicotine **11** (Supplementary Fig. 5). We propose that (*R, S*)-dihydronicotine glucoside (**5b**) can either decompose into Glc-**11** or oxidise into (*R, S*)−**13**. The former can occur

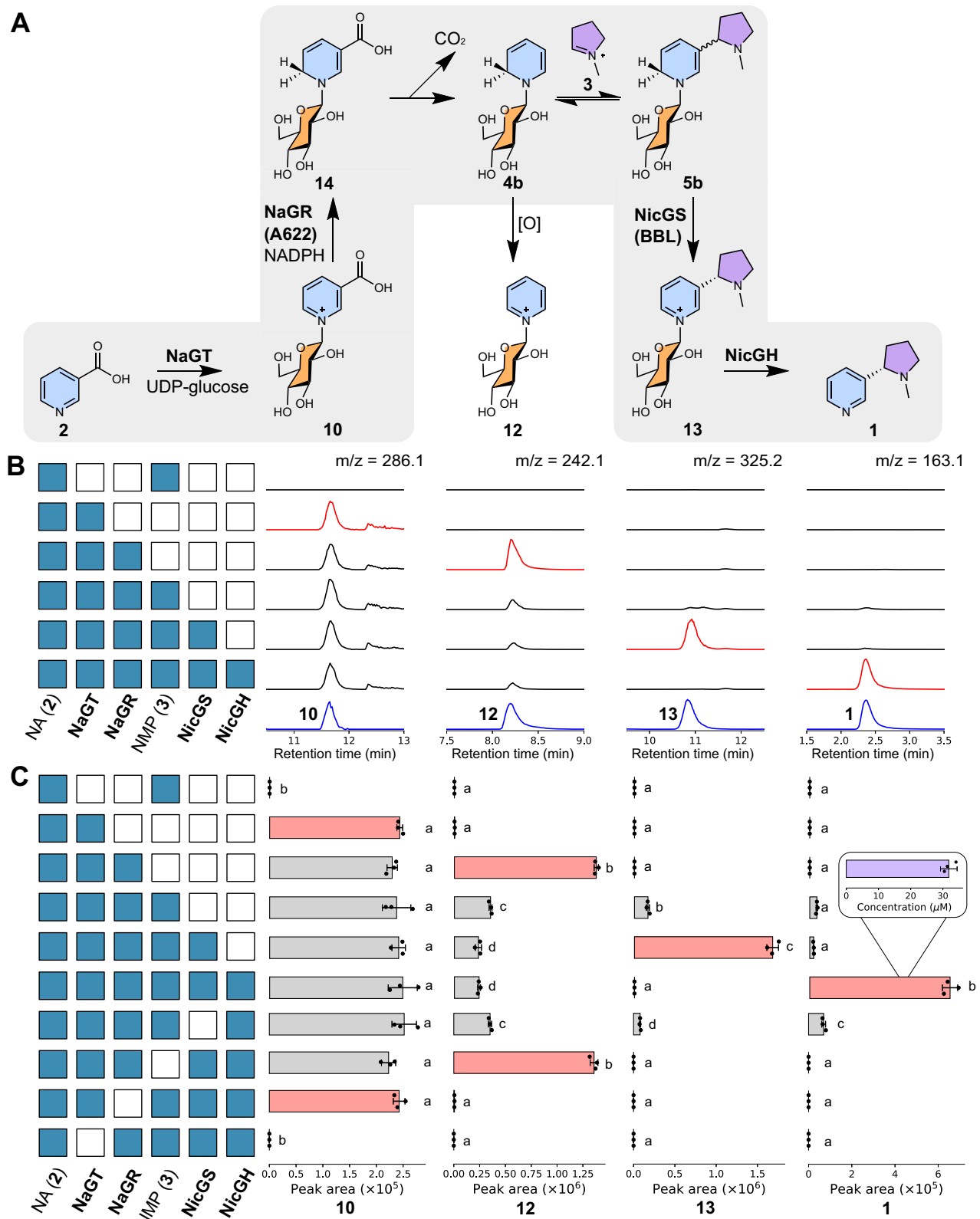

**Fig. 3 | In vitro reconstitution of nicotine biosynthesis. A** Proposed chemical transformations in nicotine biosynthesis reconstitution. **B** Products of in vitro cascades. Each row corresponds to an in vitro reaction, with the matrix showing presence/absence (blue/white) of reaction components, alongside UDP-Glc and NADPH. Each column shows extracted ion chromatograms (EICs, $m/z \pm 0.15$). Blue chromatograms are chemically verified standards, scaled for clarity. Red chromatograms highlight key products. MS$^2$ data is available in Supplementary Fig. 3. **C** Quantification of in vitro cascades. Each row of bars corresponds to an in vitro reaction, with the matrix showing presence/absence (blue/white) of reaction components. Each

column of bars shows the peak area of EICs ($m/z \pm 0.15$) corresponding to the masses of the chemical depicted above (left-to-right: **10**, **12**, **13**, **1**). Compound identity was validated by standard retention time, and MS$^2$ spectra (Supplementary Fig. 3). Red bars highlight key products. The letters above bars are significant groupings of peak area, determined per chemical ($p < 0.05$, Tukey post-hoc test, see Supplementary Data 5). The concentration of nicotine produced (from 1 mM of **2**) was determined via an external standard curve; error bars show standard error (inset). Bars show mean EIC peak area ($n = 3$); points are individual replicates; error bars show standard deviation. Source data are provided as a Source Data file.

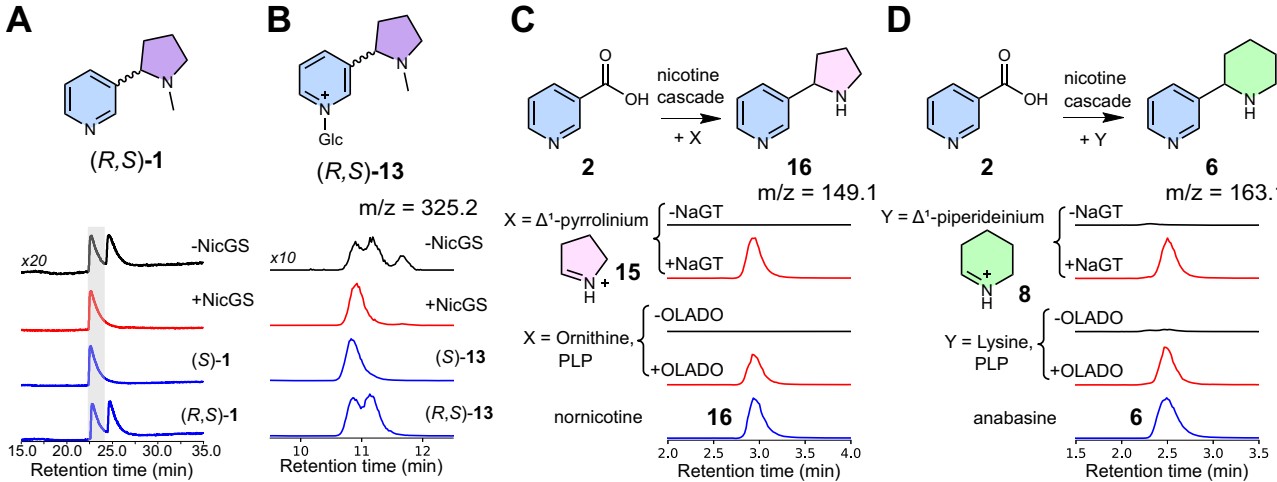

**Fig. 4 | Stereoselectivity and substrate scope of nicotine biosynthetic cascade.**
**A** Stereoselective formation of (*S*)-nicotine (**1**). Nicotine formation was achieved through one-pot enzymatic cascade (**2**, **3**, NaGT, NaGR, NicGH) with and without NicGS detected by chiral HPLC (260 nm). Signal intensity of peaks and standards (blue) was scaled for comparison. **B** Induction of stereoselectivity by NicGS. EIC (*m/z* 325.2 ± 0.15) showing (*R*, *S*)-nicotine glucoside (**13**) formation from **2** with NaGT, NaGR and NMP, with or without NicGS. Chemically verified standards are blue. Signal intensity was scaled for comparison. **C** Formation of nornicotine (**16**) and **D** anabasine (**6**) via the nicotine biosynthetic cascade (**2**, NaGT, NaGR, NicGS, NicGH) with **15** or **8**, respectively, either synthesized or formed enzymatically by OLADO.

through a series of tautomerisation steps forming Glc-**11** and be followed by hydrolysis to yield **11** (Supplementary Fig. 5D). Notably, NaGR failed to catalyse the reduction of pyridine glucoside (**12**), confirming that this is a dead-end shunt metabolite (Supplementary Fig. 6A).

Inclusion of BBLa led to a significant decrease in **12**, increase in **13** and rendered the reaction diastereoselective, with the (*S*)−**13** isomer in excess (Figs. 3 and 4B), reflected in the exclusive production of the (*S*)−**1** enantiomer in the full pathway reconstruction (Fig. 4A). In the cascades, the peaks corresponding to **11** and Glc-**11** are absent in the presence of BBLa; yet Glc-**11** is not a substrate of the enzyme, as it cannot be converted into **13** by BBLa (Supplementary Fig. 6B). This suggests that BBLa's substrate is **5b**, which is upstream of Glc-**11**: conversion of **5b** to (*S*)−**13** prevents accumulation of Glc-**11** (Supplementary Fig. 5D). The presence of **11** in the absence of BBLa matches plant silencing experiments of the BBLs[30,31]. These results redefine BBLa as (*S*)-nicotine glucoside synthase (NicGS).

The final step of the pathway is the deglucosylation of (*S*)−**13** to form (*S*)−**1**, which can be catalyzed by β-GD1, now redefined as nicotine glucoside hydrolase (NicGH) (Fig. 3). The enzyme NicGH can catalyze the deglucosylation of both diastereomers of **13** but cannot accept nicotinic acid *N*-glucoside **10**, dihydrometanicotine glucoside Glc-**11** or pyridine glucoside **12** as substrates (Supplementary Fig. 7).

### In vitro biosynthesis of alternative tobacco alkaloids
We anticipated that we could exchange *N*-methylpyrrolinium (**3**) for different electrophiles to form related pyridine alkaloids. We demonstrated this with Δ¹-pyrrolinium (**15**) and Δ¹-piperideinium (**8**), to form the natural products nornicotine (**16**) and anabasine (**6**), respectively (Fig. 4C, D). We generated the electrophiles via two approaches, chemically and via in situ oxidative decarboxylation from ornithine and lysine respectively, catalysed by a recently discovered ornithine/lysine/arginine decarboxylase-oxidase (OLADO) from *N. tabacum*[42].

### In vitro assays with isotopically labelled nicotinic acid
Feeding experiments with isotopically labelled precursors have been crucial in establishing atomic-level information about nicotine biosynthesis, such as the loss of the C6 hydrogen from nicotinic acid (Fig. 1)[17–19,31]. We reconstructed the four-enzyme nicotine biosynthesis cascade using isotopically labelled nicotinic acid-(*ring*-d$_4$) (**2**-d$_4$) as a substrate (Fig. 5 and Supplementary Fig. 8; Supplementary Data 6).

With the full cascade, we observed accumulation of the major product **1**-d$_3$ (Fig. 5B, row *i*). A similar labelling pattern is observed for nicotine glucoside (**13**) in the absence of NicGH (Fig. 5B, row *ii*). In contrast, when NicGS (BBLa) was absent, then not only is the yield of **1** reduced, but the labelling pattern of **1** is changed, with **1**-d$_4$ now the major isotopologue (Fig. 5B, row *iii*). This suggests that NicGS is responsible for the selective formation of **1**-d$_3$ in the full cascade.

To pinpoint exactly which deuterium was being exchanged, we used the singly labelled nicotinic acid-d⁶ (**2**-d⁶). With NicGS present, we saw formation of the major product **1**-d$_0$; without NicGS, **1**-d$_1$ was the major product (Fig. 5C). This verifies that the isotope exchange determined by NicGS occurs at position C6 (Fig. 5C). This labelling pattern, where the C6 deuterium from labelled **2** is lost in the product **1**, is consistent with a cascade where NaGR (A622) catalyses reduction of **2**, transferring an unlabelled hydride (i.e. a protide) from NADPH onto C6, followed by the NicGS (BBLa) catalysing specific removal of the deuteride from the intermediate 1,2-dihydronicotine glucoside (**5b**) (Fig. 5A). The overall effect is a swapping of the deuteride for protide at C6, consistent with previous observations[17–19,31].

In the absence of NicGS, the ratio of isotopologues is not equal, but reversed, with the dominant isotopologue products being the fully labelled **1**-d$_1$ or **1**-d$_4$ from **2**-d⁶ or **2**-d$_4$ respectively (Fig. 5B, row *iii*, and Fig. 5C). This indicates that the non-enzymatic oxidation of **5b** favours the abstraction of the unlabelled protide (introduced by NaGR from NADPH) rather than the deuteride (from the labelled substrate **2**). This bias occurs due to the kinetic isotope effect, which renders carbon-deuterium bonds stronger than carbon-protium bonds, so the latter are broken more readily in the non-enzymatic reaction, where selectivity is determined by bond energy rather than spatial context.

When **2**-d$_4$ is used a substrate, the NicGS (BBLa) also appears to have an effect on the labelling of pyridine glucoside **12**, which accumulates in the absence of *N*-methylpyrrolinium (**3**). Without NicGS (BBLa), the ratio of fully labelled **12**-d$_4$ to partially labelled **12**-d$_3$ is significantly greater than when NicGS (BBLa) is present (Fig. 5B, rows *iv* and *v*, Supplementary Data 6). This implies that NicGS can bind and catalyse the oxidation **4b** as well as **5b**. Cascade formation of nornicotine-d$_3$ (**15**-d$_3$) and anabasine-d$_3$ (**6**-d$_3$) from **2**-d$_4$ confirmed they are also formed via NicGS (BBLa)-catalysed oxidation (Fig. 5D, E).

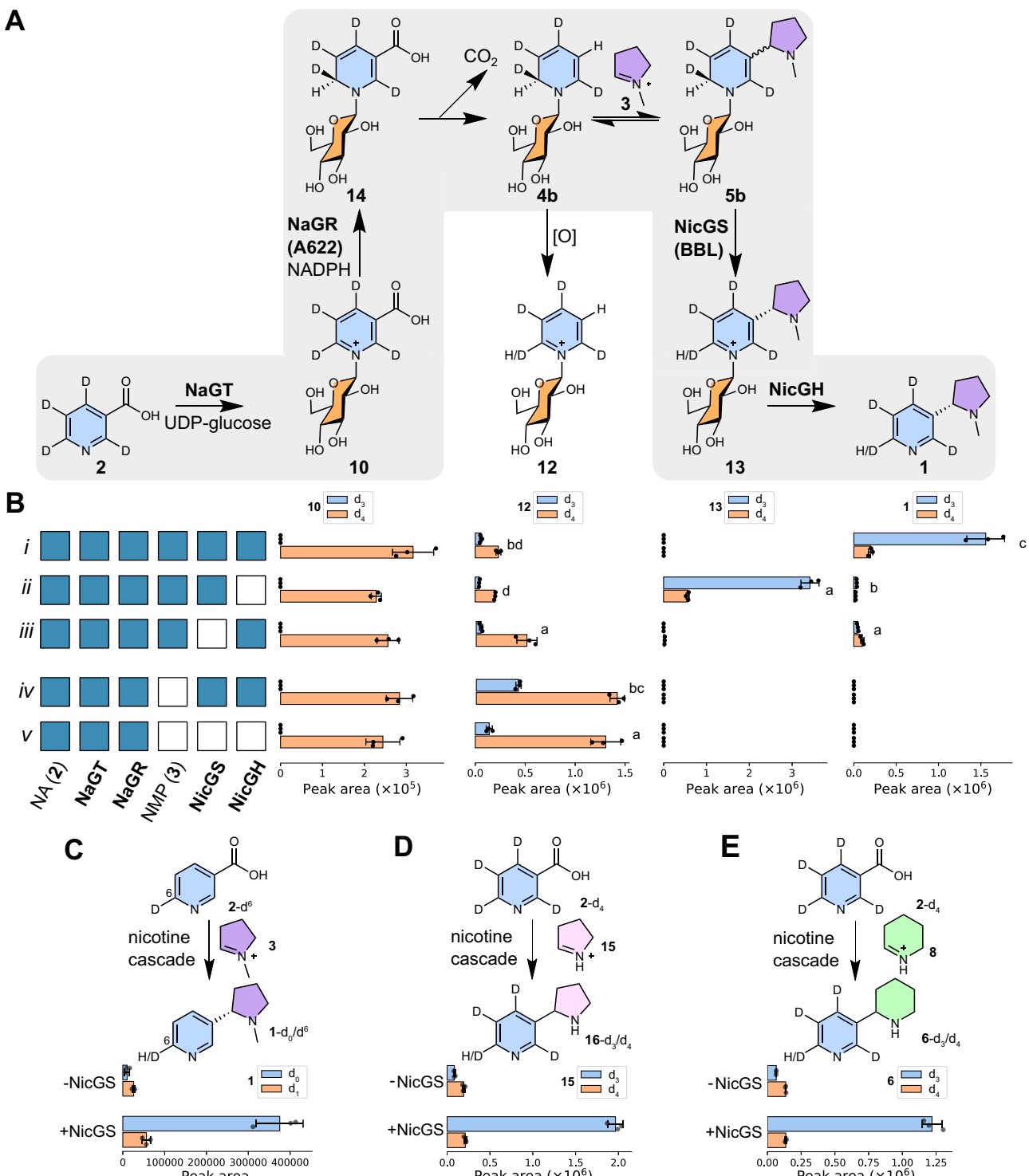

**Fig. 5 | In vitro enzymatic cascade with labelled nicotinic acid. A** Schematic of reaction cascade with nicotinic acid-($ring$-$d_4$) (**2**-$d_4$). **B** Selected in vitro cascades with **2**-$d_4$ substrate. Bar charts show EIC peak areas of product (from L-to-R: **10, 12, 13, 1**) isotopologues (all $m/z \pm 0.15$ except for $d_3$-**12** which was $m/z \pm 0.5$ due to peak shape distortion). Reactions performed in triplicate, bars show mean peak area, points show individual measurements, and error bars show standard deviation. Each row corresponds to an in vitro reaction, with the matrix showing presence/absence (blue/white) of reaction components. Letters above bars are significant groupings of $d_3$/$d_4$ isotopologue ratios, determined per chemical across all reaction combinations ($p < 0.05$, Tukey's HSD test, Supplementary Data 6, see Supplementary Fig. 8 for all reactions). **C**–**E** Conversion of nicotinic acid (**2**) isotopologues by the nicotine biosynthetic cascade (NaGT, NaGR, NicGH, NADPH and UDP-Glc) with and without NicGS. Bar charts show peak areas of products from EICs. Reactions were performed in triplicate (points); bars show mean peak area; error bars show standard deviation. Source data are provided as a Source Data file.

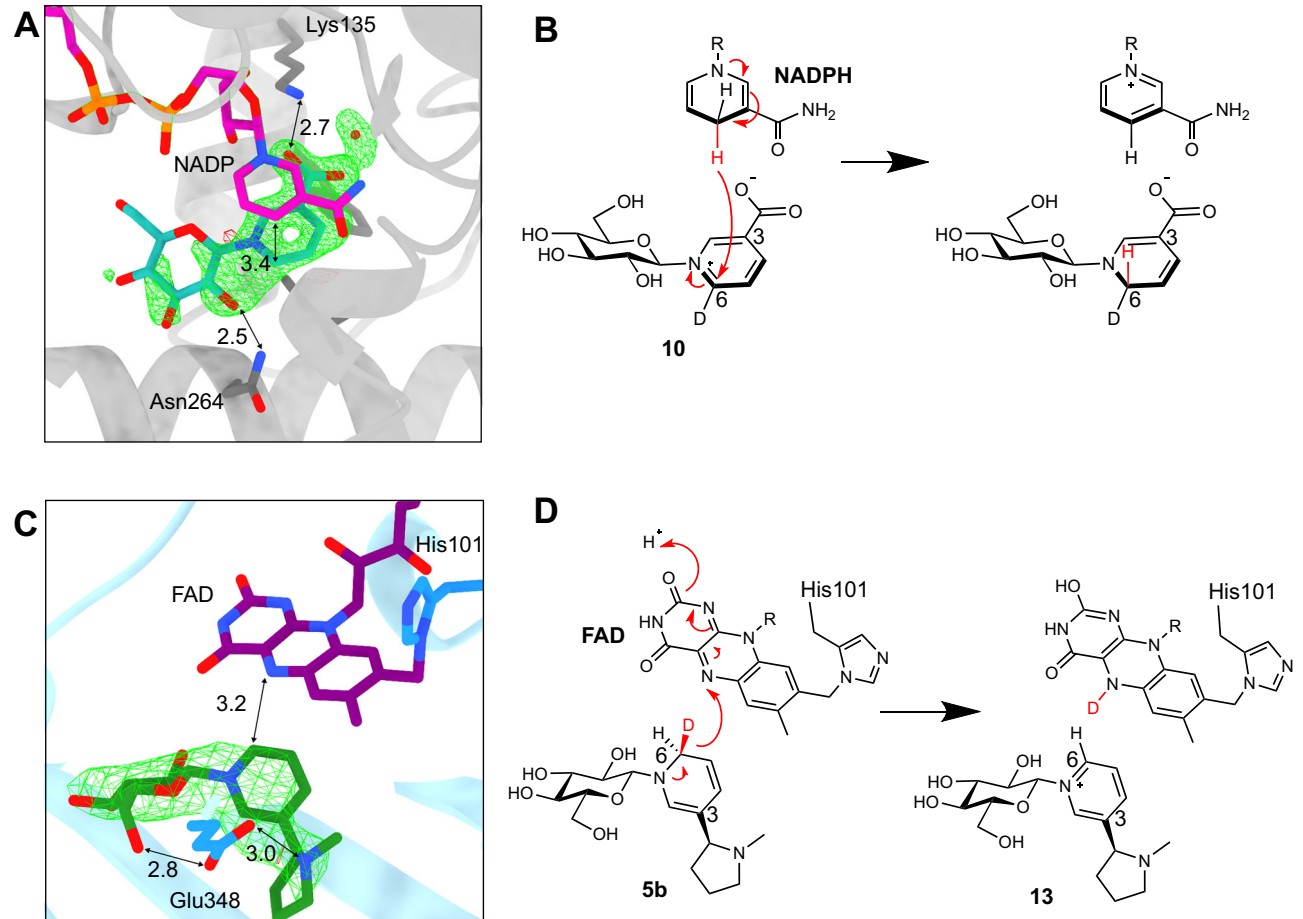

**Fig. 6 | X-ray crystal structures of NaGR (A622) and NicGS (BBLa). A** The crystal structure of NaGR bound to NADP⁺ and nicotinic acid *N*-glucoside (green net shows unrefined ligand density Fo-Fc at 3 σ). **B** Proposed mechanism of NaGR: NADPH reduces nicotinic acid *N*-glucoside-d⁶ on the *re*-face, adding a hydride in the pro-*R* position. **C** The crystal structure of NicGS with covalently bound FAD and ligand (*S*)-nicotine glucoside (green net shows unrefined ligand density Fo-Fc at 3 σ). **D** Proposed mechanism of NicGS, abstracting the pro-*S* hydride from dihydronicotine glucoside-d⁶ to yield nicotine glucoside.

## Structure of NaGR (A622)

To shed light on the mechanisms of NaGR (A622) and NicGS (BBLa), we determined their structures in complex with co-factors and relevant ligands using X-ray crystallography (Supplementary Data 7). In the case of NaGR (A622), we obtained a 1.3 Å resolution crystal structure in complex with NADP⁺ and nicotinic acid *N*-glucoside (**10**) (9RDD) (Supplementary Fig. 9). NaGR (A622) is most similar to dehydrogenases of the isoflavone reductase family (*e.g.* 2GAS, 57% sequence identity)[43], with the cofactor NADP⁺ bound between the Rossmann fold and the substrate binding domain. Omit density corresponding to the ligand nicotinic acid *N*-glucoside **10** was observed adjacent to the cofactor, with the C6 atom of the pyridine ring poised to receive hydride from C4 of the nicotinamide ring of the cofactor at a distance of 3.4 Å (Fig. 6A). This matches the regio- and stereoselectivity of the hydride transfer from NADPH onto the *re*-face of **10** at C-6, and into the pro-*R* position (Fig. 6B). The carboxylate is bound by the side chain of Lys135. The sugar density was not complete, but an interaction was observed between the side chain of Asn264 and the C2 hydroxyl of the sugar.

## Structure of NicGS (BBLa)

We obtained 2.5 Å resolution structures of NicGS (BBLa) in complex with FAD (9RUW), and with both FAD and the putative reaction product (*S*)-nicotine glucoside (**13**) (9RDR) (Supplementary Data 7 and Fig. 6C; Supplementary Fig. 9). NicGS is structurally most similar to the

reticuline oxidase/berberine bridge enzyme from *Eschscholzia californica* (3D2D, 44.6% amino acid identity)[44], but the FAD is not bi-covalently bound as in the case of 3D2D, with C166 replaced by G165 in NicGS. Omit density was observed adjacent to the FAD density that was successfully modelled as (*S*)-**13** (Fig. 6C). The C-6 position of **13** is 3.2 Å from the reactive nitrogen of the FAD cofactor, with the *si*-face of the pyridine ring facing the FAD. This proximity indicates that the C6 pro-*S* hydride of the proposed dihydronicotine glucoside (**5b**) substrate reduces the FAD cofactor (Fig. 6D). The side chain of E348, not conserved in 3D2D, has a role in substrate recognition, interacting with the nitrogen of the pyrrolidine ring and also the exocyclic C6 hydroxyl of **13** at distances of 3.0 and 2.8 Å respectively. In the absence of ligand binding (9RUW) the loop containing this residue is unstructured (Supplementary Fig. 9). Overall, the structures of NaGR (A622) and NicGS (BBLa), with bound glucoside substrates or products lend strong support to our pathway proposal. Furthermore, the atomic details provide a structural/mechanistic validation of isotope labelling experiments, where NicGS (BBLa) abstracts the opposite hydride to that added by NaGR (A622).

## *In planta* pathway reconstitution

To demonstrate that the pathway assembled in vitro could be operational *in planta*, we reconstituted nicotine biosynthesis in leaves of *N. benthamiana* via transient gene expression (Fig. 7 and Supplementary

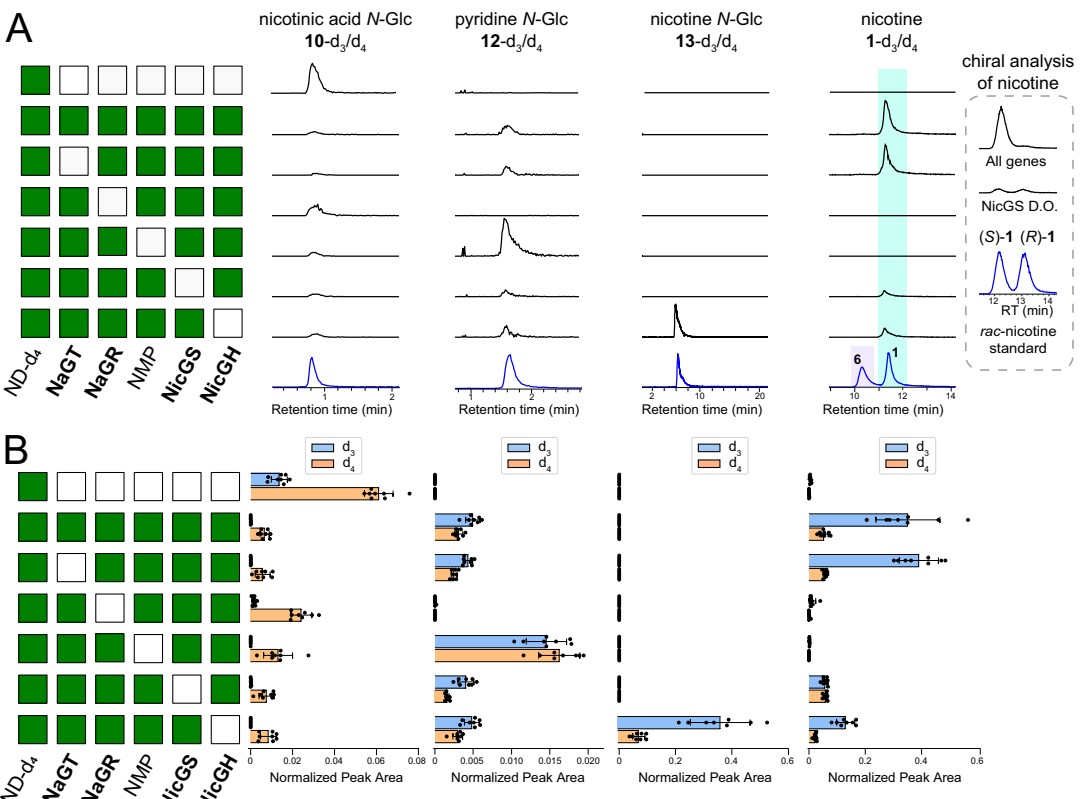

**Fig. 7 | *In planta* reconstitution of nicotine biosynthesis.** Leaves of *N. benthamiana* were agroinfiltrated with a combination of nicotine biosynthesis genes (day 0), fed with nicotinamide-d₄ (ND-d₄) at day 4, and harvested for metabolite analysis at day 6. The gene combination producing the *N*-methylpyrrolinium (ODC, PMT and MPO) is abbreviated as NMP. **A** Accumulation of the labelled compounds is indicated at the top (d₃ and d₄ versions together). Each row corresponds to the gene combination shown in the matrix on the left (green/white indicates presence/absence). Traces are representative extracted ion chromatograms (EICs, $m/z \pm 0.005$) corresponding to the labelled compounds (black) in comparison to chemically synthesized or commercial standards (blue) (see Supplementary Fig. 12

for MS² spectra). Signal intensities are normalized to the largest peak on each column; standards were rescaled to match the largest peak. The nicotine and anabasine standards were analysed as a mixed sample. The insert on the right shows the chiral analysis of nicotine in the presence of all genes and in the NicGS drop-out (D.O.). **B** Individual quantification of d₃ and d₄ isotopologues corresponding to the compounds and gene combinations shown in (**A**). Bars show mean EIC peak area ($n = 8$); points are individual replicates; error bars show standard deviation (see statistics in Supplementary Data 11). Source data are provided as a Source Data file.

Data 8)[45]. Whilst *N. benthamiana* is capable of producing nicotine, we were able to independently study the reconstituted pathway as the key endogenous genes *A622* and *BBL* are not expressed in leaves (Supplementary Fig. 10 and Supplementary Data 9)[46]. Furthermore, we could distinguish between the native nicotine transported into the leaves and the nicotine produced in situ by feeding the engineered leaves with a labelled precursor (nicotinamide-d₄) and analysing labelled nicotine.

To produce *N*-methylpyrrolinium (**3**), we co-expressed the known tobacco enzymes ODC, PMT and MPO (Supplementary Data 1 and 8)[47–49]. When the entire pathway was expressed, we observed substantial production of (*S*)-nicotine-d₃ ((*S*)-**1**-d₃) (Fig. 7 and Supplementary Fig. 12), validating our in vitro observations. Additionally, stepwise reconstruction of the pathway mirrored results from the in vitro stepwise reconstruction, including production of anabasine-d₃ (**6**-d₃) enabled by co-expression of OLADO in place of the enzymes forming **3** (Supplementary Fig. 11 and Supplementary Data 10).

We then conducted a drop-out experiment to validate the role of the individual components (Fig. 7 and Supplementary Data 11). Drop-out of NicGH (β-GD1) led to a decrease in labelled nicotine (**1**) and accumulation of labelled nicotine glucoside (**13**), which we did not detect in any other combination. Dropping out NicGS (BBLa/b), resulted in a similar reduction in labelled nicotine but the residual nicotine was racemic (Fig. 7A). Omission of the *N*-

methylpyrrolinium forming enzymes eliminated nicotine production and led to the accumulation of labelled pyridine glucoside (**12**). Dropping out NaGR (A622) abolished production of labelled pyridine glucoside (**12**), blocking the pathway at nicotinic acid *N*-glucoside (**10**). It was not possible to confirm the *in planta* activity of NaGT (UGT1), as control leaves fed with labelled precursor accumulated labelled **10**, likely due to high leaf expression of the native *UGT1* homolog (Supplementary Fig. 10). Similar to the in vitro experiments, the *in planta* experiments were able to ascribe the loss of deuterium to NicGS (BBLa/b) (Fig. 7 and Supplementary Fig. 11; Supplementary Data 10 and 11).

## Metabolites in wild-type and mutant plants

We further probed our pathway hypothesis by searching for proposed intermediates or shunt metabolites in wild-type roots of *N. tabacum* and *N. benthamiana*. We were able to detect nicotine glucoside (**13**) in *N. tabacum* and pyridine glucoside (**12**) in both *N. tabacum* and *N. benthamiana* (Fig. 8 and Supplementary Figs. 13; S14). We also examined roots of existing *N. benthamiana* knockout lines in NaGR (*A622/A622L*)[29] and NicGS (*BBLa/b/c/d/d'*)[31]. The NaGR (A622) knockout accumulated nicotinic acid *N*-glucoside (**10**) but not pyridine glucoside (**12**), supporting the identity of **12** as a shunt product of NaGR (A622) (Fig. 8A). The NicGS (BBL) knockout accumulated pyridine glucoside (**12**) and nicotine glucoside (**10**), similar to wildtype *N. tabacum* roots (Fig. 8A).

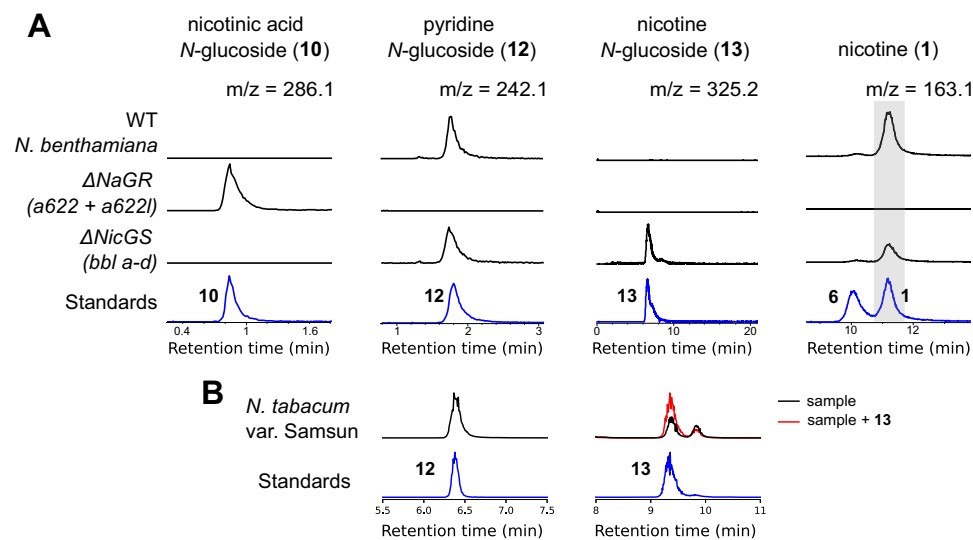

**Fig. 8 | Identification of pathway intermediates and shunt products in *Nicotiana* roots. A** Metabolite analysis of roots from *N. benthamiana* wild type, *ΔNaGR*[29] and *ΔNicGS*[31] lines (black) in comparison to standards (blue) (see MS² verification in Supplementary Fig. 13). Traces are representative extracted ion chromatograms (EICs, $m/z \pm 0.005$) and were normalized to the largest peak in each column; standards were re-scaled to match the largest peak. **B** Metabolite analysis of roots from *N. tabacum* var Samsun (black), compared to standards (blue) (see MS² verification in Supplementary Fig. 14). The red trace is root extract sample spiked with 10 µM nicotine *N*-glucoside (**13**) to verify peak identity. Traces are representative EICs ($m/z \pm 005$); standards were re-scaled to match the corresponding peaks in the *N. tabacum* samples.

## Discussion

It is remarkable that, despite the massive global use of tobacco, its economic importance, and the volume of investigation into nicotine biosynthesis, key steps in its formation have remained enigmatic. Here, we have reconstituted, in vitro and *in planta*, a minimal nicotine biosynthetic cascade, forming (S)-nicotine (S)-(**1**) from nicotinic acid (**2**) and *N*-methylpyrrolinium (**3**). It consists of four enzymes: NaGT, NaGR, NicGS, and NicGH.

Whilst we elected to test only a single representative paralog for each enzyme type in our in vitro assays, there is considerable evidence to suggest that the highly similar paralogs (or homeologs, i.e., paralogs derived from whole genome duplications) are functionally equivalent. NaGR (A622) has 95.5% protein sequence identity to its homeolog A622L and has similar expression patterns (Supplementary Data 2). Genome editing experiments show that both homeologs must be knocked out to guarantee ultra-low nicotine content in mutant tobacco lines[28,50]. Similarly, BBLa (NicGS) and BBLb are homeologs with 92.8% protein sequence identity, similar expression patterns (Fig. 2C), and genome editing experiments implicate both genes in the formation of (S)-nicotine[34]. The newly described NicGH (*β-GD1*) shares 91.8% with its homeolog *β-GD2* and 83.6% protein sequence identity with its non-syntenic paralog *β-GD3*; all genes have similar expression patterns across tissues (Fig. 2C). During the revision of this manuscript, a report was published describing the systematic assessment of *β*-glucosidase genes in *N. tabacum*[51]. It revealed that the expression of NicGH (*β-GD1*) and its paralogs *β-GD2* and *β-GD3* (named NtBGLU51, 50 and 49, respectively in the *β-GD* manuscript) were all strongly induced by methyl jasmonate, supporting their role in defence response.

Cryptic biosynthetic steps cannot be predicted based on the structures of precursors or products, making them a bottleneck in pathway elucidation. In nicotine biosynthesis, NaGT and NicGH are responsible for a cryptic glucosylation, which acts like an activating group, increasing the electrophilicity of nicotinic acid (**2**) and making it susceptible to reduction by NADPH. Glucosylation steps are common in natural product biosynthesis, but the use of the glucoside in nicotine biosynthesis as an activating group is distinct from other known examples of glucosylation. Natural product glucose conjugates are usually seen in the context of detoxification, allowing the host plant to store otherwise highly toxic molecules as inactivated glucosides, ready for future activation by deglucosylation[52]. For example, the benzoxazinoids DIBOA and DIMBOA are produced as defense compounds in grasses. These are sequestered in the plant vacuole as inactivated glucosides, which can be released by a specific glycosyl hydrolase for defense-related purposes[53]. This kind of conditional detoxification/activation is also seen for cyanogenic glucosides e.g. in sorghum[54] and in the regulation of brassinosteroid and cytokinin hormone homeostasis[55].

In plant metabolism, glucose esters are commonly used as acyl donors. This involves a glucose moiety being added to a carboxylic acid, and the glucose ester product undergoes transesterification, a step catalysed by serine carboxypeptidase-like acyltransferases (SCPL-ATs)[56]. This is analogous to coenzyme A thioesters, which are acyl donors that undergo transesterification catalysed by BAHD-acyltransferases[56]. Glucose ester intermediates feature in the biosynthesis of sinapate (sinapoylcholine) and the UV-protectant sinapoylmalate from sinapic acid via the activated sinapoylglucose ester intermediate in Brassicaceae[57], as well as in hormone homeostasis, forming IAA-myo-inositol conjugates via indoleacetic acid-glucoside[58]. Interestingly, a glucoside ester is an intermediate in tropane alkaloid biosynthesis, a pathway from Solanaceae species that shares many features with nicotine biosynthesis. In particular, a UGT glucosylates phenyllactate to phenyllactylglucoside, enabling SCPL-AT catalysed transesterification to littorine with tropine[59]. The glucosyl group in glucose esters acts to activate the carboxylic acid by forming an ester and becoming a favourable leaving group, with glucose being a product of the transesterification reaction. This is distinct from the proposed glucose activation in nicotine biosynthesis, where the glucose is not a direct product of the core reaction but instead modulates the activity of intermediates and requires a dedicated removal step: NicGH catalysed hydrolysis.

A cryptic glycosylation was recently described in Solanaceae steroidal glycoalkaloid biosynthesis, involving the cholesterol glucuronosyl transferase GAME15[60,61]. A cryptic glucosylation was also discovered in the biosynthesis of the yeast antibiotic anisomycin[62]. In both of these examples, it appears that the glycosyl moiety acts mostly as a protecting group/recognition handle for the downstream

**Fig. 9 | Putative activities of NicGS (BBL).** Possible mechanistic routes towards (S)−1 from **10**. The blue background shows key NaGR and NicGH reactions and major products. The grey background shows possible reactions that are either non-enzymatically or catalysed by NicGS. Only compounds with blue or white background have been directly detected. To induce the stereoselective outcome, NicGS may operate through: dynamic kinetic resolution (catalysing just purple arrows and allowing other intermediates to non-enzymatically equilibrate), epimerisation of **5b** via a ring opened intermediate (catalysing red and purple arrows), catalysis of the stereoselective Mannich reaction between **3** and **10** prior to oxidation, reversal of the non-desired Mannich reaction (opening (R)-**5b** into **3** and **10**). Blue arrows show the predicted mechanism of Glc−**11** formation.

biosynthetic enzymes, rather than having an activation effect as we propose for nicotine biosynthesis.

As well as modulating reactivity, the detoxification effect of glucoside-conjugates in plants can be attributed to driving vacuolar localisation of metabolites. This may also be true for nicotine biosynthesis, given the known subcellular localisation of NaGR (A622) to the cytosol[32], and NicGS (BBL) to the vacuole[30], and also given that nicotine accumulates in vacuoles of tobacco root cells during biosynthesis[38]. The predicted N-terminal signal peptide and C-terminal vacuolar sorting signal of NicGH strongly suggest it will also localise to the vacuole (Supplementary Fig. 1). The recent manuscript reporting tobacco β-glucosidases included an experiment where GFP was fused to the C-terminus of β-GD2 (NtBGLU50) and the fusion protein was transiently expressed in *N. benthamiana*. Surprisingly, the protein appeared to localise to the endoplasmic reticulum, despite its predicted vacuolar localisation sequence. Further investigation of the subcellular localisation of β-GDs is required for a conclusive outcome.

The division of the nicotine biosynthetic cascade across subcellular compartments—the cytosol and the vacuole—does not support the hypothesis that the cascade can operate as a structurally defined multi-enzyme complex. Furthermore, the reported 'nicotine synthase' enzyme isolated from tobacco roots by Friesen and Leete in 1990 was obtained with minimal purification: just filtration, desalting and crude ammonium sulfate protein fractionation, with activity observed in the 25–75% fraction[16]. Therefore, whilst the nicotine biosynthetic cascade enzymes (NaGT, NaGR, NicGS, NicGH) and their cofactors must have been present and soluble in this broad fraction, without further purification or alternative evidence, there is no reason to presume that the cascade acts as a 'nicotine synthase' enzyme complex.

The substrate for NaGR (A622) is nicotinic acid N-glucoside (**10**), which is consistent with its accumulation when *A622* is silenced or knocked out[26]. Previous attempts to characterise this reaction through spectrophotometry, typical for NADPH-dependent reactions, may have been hampered by overlapping absorbance spectra in the NADPH

substrate and the 1,2-dihydropyridine glucoside (**4b**) product. Furthermore, **4b** appears unstable in vitro: here, its presence is inferred through the identification of pyridine glucoside (**12**) or through the reaction outcome in the presence of an electrophile. The cascade operates stereoselectively, with formation of (S)-enantiomer of **1** dominating both in vitro and *in planta* (Figs. 4A and 7A). NicGS (BBLa) is responsible for determining the stereoselectivity of the pathway, as we previously showed *in planta*[31]. The crystal structure of NicGS bound to (S)-nicotine glucoside ((S)-**13**) indicates it catalyses the diastereoselective oxidation of dihydronicotine glucoside (**5b**), but it is unclear whether it also catalyses a stereoselective Mannich reaction or a dynamic kinetic resolution process. The accumulation of the shunt product pyridine glucoside **12** is inhibited in the presence of **3**, even in the absence of NicGS, indicating facile non-enzymatic coupling between **4b** and **3**. However, the decreased concentration of **12**, higher conversion into **13** and induction of stereoselectivity in the presence of NicGS suggest enzymatic control of this step, as does the observation that NicGS can catalyse oxidation of **4b**, which would match stepwise binding of **4b** and **3**. Further work is required to resolve the intricacies of the NicGS mechanism, given the complexity of the enzyme cascade (Fig. 9). The modularity of the cascade with respect to the electrophile has been demonstrated by the formation of anabasine and nornicotine, and suggests the system could form the basis of a biocatalytic route towards diverse functionalised pyridines. New genetic understanding of nicotine biosynthesis also has the potential to inform engineering of *Nicotiana* sp., including the elimination or rerouting of alkaloid biosynthesis for new synthetic biology chassis. This work resolves a centuries-old puzzle in chemistry, plant biology and biochemistry, and will lead to the development of novel biotechnological and biocatalytic tools to produce valuable chiral chemicals.

In the period between completing final revisions to this manuscript and it being accepted, an article from the group of Dapeng Li was published describing their independent discovery of enzymes involved in nicotine biosynthesis[63].

## Methods

### Expression analysis

Genes of interest were defined by assessment of the literature related to nicotine biosynthesis (Supplementary Data 1). Protein sequences were obtained via GenBank or UniProt accession numbers and used as tblastn queries against the coding sequences from the 2017 *N. tabacum* genome[37] to obtain gene names and sequences. Biosynthetic gene clusters were identified using functional annotations and RNA-seq profiles of genes neighbouring genes of interest, using JBrowse hosted on the SolGenomics[64], querying the 2017 *N. tabacum* genome[37]. Gene names were used as queries for the Plant Gene Expression Omnibus (PEO) (https://peo.ku.dk/)[40]. The PEO *N. tabacum* dataset consists of TPMs for 824 samples over twelve tissue types. Precomputed Pearson Correlation Coefficient (PCC) values were used to identify top-correlating genes.

For gene co-expression analysis, TPMs of genes of interest were extracted from PEO and analysed in R-Studio (tidyverse and pheatmap packages). For plotting gene expression from *N. tabacum* across tissues, $\log_2(\text{TPM} + 1)$ values were grouped by Plant Ontology (i.e. tissue type), averaged, and plotted as a heatmap, with Z-score normalisation across tissue types. For expression correlation, an all-by-all PCC matrix was computed (824 samples, $\log_2(\text{TPM} + 1)$), depicted as a heatmap, with genes grouped with complete-linkage hierarchical clustering on Euclidean distance. *Nicotiana benthamiana* nicotine biosynthesis genes homologs were identified by top BLASTN hit using *N. tabacum* genes as queries of the *N. benthamiana* v1.0.1 genome[64]. Gene names were used as queries for PEO and TPM values were extracted (476 samples), grouped by Plant Ontology and plotted as boxplots ($\log_2(\text{TPM} + 1)$)[40].

For assessment of gene expressions after topping, we extracted a list of differentially expressed genes (DEGs) in tobacco roots after topping, from Qin et al 2020[41]. Using the mean FPKM value at $t = 0$ we converted $\log_2$ fold change values to FPKM values across the time course and then calculated PCCs with A622 (Nitab4.5_0000884g0010) expression. Genes were filtered by annotation 'UDP-glucosyltransferase' and $\text{PCC}_{A622} > 0.85$. Top hits were assessed for their overall expression level and used as a query in PEO.

### Protein purification

Codon-optimised NaGR (A622, UniProt Accession: IFRH_TOBAC), NaGT (UGT1, 2017 *N. tabacum* genome: Nitab4.5_0006222g0020[37]) were expressed using the pET28a(+) vector as C-terminal (NaGR) or N-terminal (NaGT) His-tagged proteins, using *E. coli* B834 or SoluBL21, respectively. NaGR was expressed as a Δ5 N-terminal truncated form. Proteins were purified using nickel affinity chromatography followed by size-exclusion chromatography. NicGS (BBLa, 2017 *N. tabacum* genome: Nitab4.5_0006307g0010) and NicGH (*β*-GD1, 2017 *N. tabacum* genome: Nitab4.5_0000884g0020) were expressed as N-terminal His-tagged proteins using a *Komagataella phaffii* secretion system based on the pPICZα vector. Proteins were purified from the media using tangential flow filtration, followed by nickel affinity chromatography and size-exclusion chromatography. OLADO (UniProt Accession: A0A1S3ZKS9_TOBAC) was expressed using the pET28a(+) vector as an N-terminal His-tagged protein using *E. coli* SoluBL21 and purified using nickel affinity chromatography followed by buffer exchange using a PD10 column[42]. Protein concentration was estimated using a Nanodrop spectrophotometer with predicted extinction coefficients derived from Expasy ProtParam, and then stored at -70 °C. Details of sequences are available in Supplementary Data 1 and 4), further methodological details are available in the Supplementary Information.

### In vitro assays

In vitro enzyme assays and multi-enzyme cascades were carried out in 50 μL reactions, in 50 mM phosphate buffer, pH 7.4, 100 mM NaCl.

Reactions contained 1 mM nicotinic acid, 1 mM *N*-methylpyrrolinium, 0.5 mg mL$^{-1}$ NaGT (UGT), 0.5 mg mL$^{-1}$ NaGR (A622), 50 μg mL$^{-1}$ NicGS (BBLa) and 5 μg mL$^{-1}$ NicGH (*β*-GD). Other combinations were tested by removing component(s) of this mix to make different combinations of enzymes/substrates. The reactions were incubated for 16 hours at 37 °C, then quenched (200 μL, 90:10 acetonitrile: water, 20 mM ammonium formate, pH 3), centrifuged ($21,000 \times g$, 10 min) and the supernatant was transferred to vials for LC-MS analysis. The *N*-methylpyrrolinium was prepared by incubation of 1 μL γ-methylaminobutyraldehyde diethyl acetal with 49 μL 1 M HCl at 70 °C for 1 h before neutralisation with 40 μL 1 M NaOH, incubated on ice until adding to the assay on the same day. 1-Pyrroline for nornicotine assays was prepared in the same way from aminobutyraldehyde diethyl acetal. Enzyme assays were analysed by LC-MS, with Hydrophilic Interaction Liquid Chromatography (HILIC) separation using a Waters XBridge BEH Amide column (5 μm, 2.1 × 100 mm) on a Thermo Scientific Vanquish UHPLC. Detection was performed on a Thermo Scientific LTQ XL Linear Ion Trap Mass Spectrometer. Details on LC-MS analytical procedures can be found in the Supplementary Information.

### Chiral nicotine analysis of an in vitro reaction

In vitro enzyme assays for chiral analysis were carried out in 3 × 50 μL reactions for each condition, in 50 mM phosphate buffer, pH 7.4, 100 mM NaCl. Reactions contained 5 mM nicotinic acid, 5 mM *N*-methylpyrrolinium, 10 mM UDP-Glc, 2 U Glucose-6-phosphate dehydrogenase (Merck), 5 mM NADP and 10 mM glucose-6-phosphate, with 0.5 mg mL$^{-1}$ NaGT (UGT), 0.5 mg mL$^{-1}$ NaGR (A622), 50 μg mL$^{-1}$ NicGS (BBLa) and 5 μg mL$^{-1}$ NicGH (*β*-GD). Samples for chiral HPLC analysis were basified with 1 μL 1 M NaOH, then protein precipitated with 500 μL cold EtOH. After centrifugation ($21,000 \times g$, 10 min), the protein pellet was washed with a further 500 μL EtOH, the EtOH was then combined, evaporated to dryness and resuspended in 50 μL propan-2-ol and then loaded into vials for analysis. Assays with NicGS were diluted 10-fold further. Nicotine chiral analysis was carried out on an Agilent 1200 HPLC by normal phase isocratic elution using a CHIRALPAK® AD-H 250 ×4.6 mm column using 99:1 hexane: IPA at 1 mL min$^{-1}$. Nicotine peaks were detected by UV absorbance at 260 nm.

### In vitro assay statistics

Statistical analysis of in vitro reaction product formation was performed using a one-way ANOVA (R-studio, function: aov) to determine EIC peak area differences across unique reactions, followed by a Tukey's HSD post-hoc test (function: TukeyHSD), with peak area grouping assigned letters (function: multcompLetters) based on significant differences ($p < 0.05$) in pairwise comparisons. Reactions were performed in triplicate ($n = 3$). For analysis of isotopologue ratios, $d_3/d_4$ isotopologue ratios were calculated per chemical and per sample through integration and division of EIC peak areas. Samples with zero peak area for either isotopologue were removed from the analysis. Then isotopologue ratios ($n = 3$) for each chemical were compared across unique in vitro reactions, analysed by analysis of variance and Tukey's HSD as described above. To calculate the nicotine concentration of the complete cascade, a standard curve of known concentrations was run in the same batch as the samples and fitted using log-log linear regression. Model parameters were used to convert sample peak areas into concentrations. To accurately estimate the mean and standard error, we first estimated the uncertainty of individual concentrations through the regression parameter errors and residual variability, before combining these using inverse-variance weighting of the triplicates to yield a pooled mean and standard error.

### Protein crystallization

Initial screening of crystallization conditions was performed using commercially available INDEX (Hampton Research), PACT premier and CSSI/II (Molecular Dimensions) screens in 96-well sitting drop trays,

which were stored at 4 °C. Drops (300 nL) were set up with a 1:1 mixture of 12 mg/mL NaGR (A622) with precipitant mother liquor. Ligand complexes of A622 were obtained by co-crystallisation with 5 mM nicotinic acid *N*-glucoside **1**, derived from a 100 mM stock solution in $H_2O$, which was added to the NaGR (A622) solution and incubated on ice for 30 min before the trays were set up. Crystals were obtained in 0.1 M HEPES, pH 7.5 and 25 % (w/v) PEG 3350. NicGS (BBLa) protein was enzymatically deglycosylated before setting up crystallization screens. 53 µL 20 mg mL$^{-1}$ NicGS (BBLa) was incubated overnight at 37 °C with 7 µL EndoH and 7 µL GlycoBuffer 3 (New England Biolabs). After incubation, the deglycosylated BBL was used for crystallisation at 15 mg/mL. Initial screening of crystallization conditions was performed as for NaGR but at 16 °C. Initial *apo*-crystals of NicGS (BBLa) were obtained in drops containing 0.14 M LiSO$_4$, 0.1 M Bis-Tris pH 5.5, 21% PEG 3350, by co-crystallisation with 10 mM (S)-nicotine glucoside **13**, derived from a 100 mM stock solution in $H_2O$. Ligand-bound crystals were obtained by co-crystallisation as described, but with 50 mM (S)-nicotine glucoside, derived from a 1 M aqueous solution of (S)-nicotine glucoside. Crystals were harvested directly into liquid nitrogen with nylon CryoLoops™ (Hampton Research), using the mother liquor without any further cryoprotectant (NaGR and NicGS *apo*) or by addition of 10% ethylene glycol (NicGS ligand-bound) to the drop prior to fishing.

## X-ray structure elucidation

The datasets described in this report were collected at the Diamond Light Source, Didcot, Oxfordshire, U.K., on beamline I03. Data were processed and integrated using XDS[65] and scaled using SCALA[66] included in the Xia2[67] processing system. Data collection statistics are provided in Supplementary Data 7. The crystals of NaGR (A622) were obtained in space group $P1$, with one molecule in the asymmetric unit; the crystals of NicGS (BBLa) were obtained in space group $P2_12_12_1$, with two molecules in the asymmetric unit. The structures were solved by molecular replacement using MOLREP[68] with the AlphaFold structures AF-P52579-F1-v4 and AF-F1T160-F1-v4 used as the models for NaGR and NicGS, respectively (https://alphafold.ebi.ac.uk/)[69]. The structures were built and refined using iterative cycles in Coot[70] and REFMAC[71] employing local NCS restraints in the refinement cycles where relevant. The final structure of NaGR (A622) ligand complex exhibited % $R_{cryst}$ /$R_{free}$ values of 17.1/21.0; the final structures of *apo*-NicGS (BBLa) and the NicGS (BBLa) ligand complex exhibited % $R_{cryst}$ /$R_{free}$ values of 29.3/32.4 and 21.5/26.9, respectively. Refinement statistics for the structures are presented in Supplementary Data 7. The structures of NaGR (A622)-NADP+nicotinic acid *N*-glucoside, NicGS (BBLa)-FAD nicotine *N*-glucoside and NicGS (BBLa)-FAD have been deposited in the Protein Databank (PDB) with accession codes 9RDD, 9RDR and 9RUW, respectively.

## *In planta* pathway reconstruction

The genes for transient overexpression were synthesized and cloned into a pHREAC vector (Addgene plasmid #134908)[72] by Twist Bioscience (San Francisco, USA). Coding sequences of the genes of interest were obtained from the *N. tabacum* v1.0 Edwards 2017 genome (scaffold)[37] and confirmed/refined by examining mapped RNAseq data in the genome browser on the website of the Sol genomics Network[64]. A list of gene identifiers and coding sequences can be found in Supplementary Data 8. The cloned sequences of all pHREAC gene inserts were checked by Sanger sequencing and the plasmids were separately transformed into the electrocompetent *Agrobacterium tumefaciens* strain AGL1. Transient expression was performed in leaves of 4-week-old greenhouse-grown wild-type *N. benthamiana* plants according to Chuang and Franke 2022[73]. pHREAC-GFP (kindly provided by Hadrien Peyret and George Lomonossoff) was used as a control and as a placeholder in the drop-out experiment. Four days post-infiltration, leaves were reinfiltrated with 400 µM nicotinamide-d$_4$

(D4 98%; Cambridge Isotope Laboratories, Tewksbury, MA, USA) dissolved in infiltration buffer without acetosyringone. Leaf disks were harvested after an additional 2 days (6 disks/plant: 3 disks/leaf x 2 leaves per plant). The harvested samples were flash-frozen in liquid nitrogen and were stored at −70 °C until tissue homogenization using chrome balls and a TissueLyzer (Qiagen). The homogenized samples were stored at -70 °C until metabolite extraction.

## Metabolite analysis of *in planta* pathway reconstruction

Metabolites were extracted from homogenized flash-frozen plant tissue ($n = 8$) as described by Vollheyde and Dudley et al. 2023[31]. For the analysis, 75 ppm (386.2 µM) caffeine was used as the internal standard in the extraction solution and the proportion of tissue to extractant (µL) was 40 mg/100 µL. After the incubation step, samples were spun down for 5 min at 16200 × $g$ (max speed table top centrifuge) and supernatants were diluted in ultrapure water at a proportion of 1:15. Samples were stored at -70 °C after extraction until filtering and after filtering until further analysis. The methanolic plant extracts were analysed via reversed-phase LC-MS according to Vollheyde and Dudley et al. 2023[31] with an injection volume of 10 µL. Compounds were identified by comparison to commercial and synthesized standards. The chiral LC-MS analysis of methanolic plant extracts was performed according to Vollheyde and Dudley et al. 2023[31] with an injection volume of 10 µL.

Statistical analysis was performed in R Studio. For each isotopologue (D0, D3, D4), a one-way ANOVA was conducted to test for differences between gene combinations (groups = 7, $n = 8$). For the dropout, the construct containing all genes was set as the reference. For the step-wise reconstruction, the construct containing GFP only was set as the reference. Post-hoc comparisons were carried out using two-tailed Dunnett's test (multcomp package) to compare each construct directly against the reference group.

## *Nicotiana benthamina* metabolite analysis

*N. benthamiana* WT and quintuple *bbla-d´* knock-out seeds (*ΔNicGS*) were kindly provided by Nicola Patron[31]. Seeds of *N. benthamiana a622* knock-out line 3-3-1 (*ΔNaGR*) were kindly provided by Boaz Negin and Georg Jander[29]. Prior to germination, seeds of each line were surface sterilized according to Florentine et al. 2016[74]. Several seeds per line were incubated in 500 µL 1% sodium hypochlorite solution for 2 min and afterward the seeds were washed three times with 500 µL ultra-pure water. The seeds were germinated in petri dishes on three layers of filter paper wetted with ultra-pure water. The plates were sealed with Leucopore and kept in a growth cabinet under long-day conditions (16 h light, 8 h dark) at day: night temperatures of 22 °C: 21 °C, 60% humidity and 200 µmol light intensity. The germinated seedlings were transferred to soil and the plants were grown in a growth cabinet under long day conditions (16 h light, 8 h dark) at day: night temperatures of 22 °C: 21 °C, 60% humidity and 200 µmol light intensity. Leaf and root samples were harvested from about 2-month-old plants. The root material was mostly collected from the soil-pot interface. To clean the roots, the soil was washed off with water, and excess water was removed by taping the roots on tissue paper. The harvested samples were flash-frozen in liquid nitrogen and were stored at −70 °C until tissue homogenization using chrome balls and a TissueLyzer (Qiagen). The homogenized samples were stored at −70 °C until further use. The samples were extracted and analysed by LC-MS as described above.

## *Nicotiana tabacum* metabolite analysis

*Nicotiana tabacum var. Samsun* was grown in a temperature-controlled glasshouse with 21/18 (SD ± 3)°C Day/Night and 16 h photoperiod. Further details of growing conditions are in the Supplementary Information. Plants' apical and axillary growing tips were removed as soon as flower buds appeared (after about 1 month) and

these were regularly removed (monthly) until roots were harvested from 3-month-old plants and freeze-dried. 10 mg of freeze-dried root tissue was extracted into 1 mL MeOH, centrifuged (21,000 × *g*, 10 min) and loaded into LCMS vials for analysis. LC-MS analysis was carried out with Hydrophilic Interaction Liquid Chromatography (HILIC) separation on a Waters ACQUITY UPLC I-Class with detection performed on a Thermo Scientific Orbitrap Fusion™ Tribrid™ mass spectrometer. Details on LC-MS analytical procedures can be found in the Supplementary Information.

### Synthetic methods
Generally, glucoside intermediates were synthesised by glucosylation of the pyridine nitrogen using acetobromo-α-ᴅ-glucose, followed by deacetylation with aqueous HBr. Detailed synthetic methods and compound characterisation are available in the Supplementary Information.

### Reporting summary
Further information on research design is available in the Nature Portfolio Reporting Summary linked to this article.

## Data availability
Crystal structure files have been deposited on the RCSB Protein Data Bank under the accessions: NaGR *holo* (nicotinic acid-*N* glucoside) 9RDD NicGS *holo* (nicotine *N*-glucoside) 9RDR NicGS *apo* 9RUW. All other data are available in the main text, the supplementary information and from the corresponding author(s) upon request. Source data are provided with this paper.

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

## Acknowledgements

We thank: Sam Hart and Dr. Johan P. Turkenburg for assistance with X-ray data collection; Diamond Light Source Didcot UK for access to beamline IO3 under grant number mx24948; Jared Cartwright, Tony Larson and the Biosciences Technology Facility; Ed Bergstrom, Karl Heaton, Mariela González, Jack Olsen and the Centre of Excellence in

Mass Spectrometry; Matthew Davy and the NMR facility; Caragh Whitehead, Tessa Keenan, Zhouqian Jiang, Inesh Amarnath and Lachlan Waddell for preliminary work and materials; Peter Bølge Sørensen, Jason Daff and the University of York Horticulture team. We thank Hadrien Peyret and George Lomonossoff for the pHREAC-GFP plasmid, Georg Jander and Boaz Negin for the gift of the *N. benthamiana a622* line, and Nicola Patron for the *N. benthamiana* wildtype and quintuple *bbla-d´* knock-out line. B.R.L. acknowledges funding from UKRI [MR/S01862X/1 and MR/X010260/1]. B.T.W.S. acknowledges funding from BBSRC [BB/T007222/1] and F.G-F. acknowledges funding from Independent Research Fund Denmark [DFF 2035-00038B, 0136-00410B and 5281-00267B].

## Author contributions

B.T.W.S. performed chemical synthesis; protein expression, purification and crystallography; and in vitro enzyme assays, pathway reconstruction and analysis. I.M.A. performed and analysed *Nicotiana benthamiana* transient expression experiments and ran the analysis of mutant plants. K.V. designed plant expression constructs and co-supervised I.M.A. Z.I. contributed to construct design, protein purification and method development. J. L. developed methods for chiral analysis. K.S.S. and C.D.S. synthesised deuterated substrates. M.A.F. supervised the design and execution of chemical synthesis. G.G. supervised protein preparation, crystallography and contributed to the X-ray structural analysis. F.G-F. supervised the *in planta* transient expression and metabolic analysis. B.R.L. identified gene candidates, supervised the biocatalytic reconstruction and coordinated the project. All authors contributed to data interpretation and writing the manuscript.

## Competing interests

C.D.S. and K.S.S. are inventors on a pending patent related to the production of deuterated compounds. The remaining authors declare no competing interests.
