## [Transparent Peer Review file · Nature Communications]

Nicotine biosynthesis completed by cryptic activating glucosylation

Corresponding Author: Dr Benjamin Lichman

Version 0:

Reviewer comments:

Reviewer #1

(Remarks to the Author)

This paper by Schwabe et al. resolves a longstanding mystery in plant specialized metabolism by elucidating the unknown terminal steps of biosynthetic pathway to nicotine in *Nicotiana*. Although “nicotine synthase” activity had been demonstrated decades ago using crude enzyme preparations, the molecular identity and mechanistic basis of this activity remained unknown. In this paper, the authors reconstitute nicotine biosynthesis *in vitro* and *in planta* using four defined enzymes and, also provide high-resolution crystal structures of two key catalytic components that illuminate the underlying enzymatic mechanism at the atomic level. Together, the biochemical reconstitution, isotope-labeling experiments, structural biology, and plant validation represent a novel and significant advance in plant natural product biochemistry. The work is carefully designed and well executed, and the findings have clear translational implications for metabolic engineering, synthetic biology, and biocatalysis of pyridine alkaloids.

1. My main comment is the use of the term “nicotine synthase.” In the historical literature, this term originated from crude biochemical activity assays, without knowledge of the molecular nature of the catalyst. Conventionally, “synthase” implies either a single enzyme or a defined enzyme complex. In the present study, four enzymes are collectively described as nicotine synthase, but it remains unclear whether these proteins form a stable complex, a transient metabolon, or act entirely independently.

The authors could address this point experimentally by testing whether the recombinant enzymes associate with one another, for example via co-expression followed by size-exclusion chromatography, native PAGE, or pull-down assays. Demonstrating a stable or semi-stable complex would strengthen the rationale for using the term “nicotine synthase,” particularly given the instability of proposed intermediates and the potential benefit of substrate channeling to enhance catalytic efficiency and suppress side reactions. If no physical association exists, I suggest the authors refrain from referring to the four enzymes collectively as “nicotine synthase,” and instead describe them as a multi-enzyme biosynthetic cascade/pathway.

2. I encourage the authors to determine and present the subcellular localization of the enzymes in *Nicotiana*. Such data would help corroborate the possibility of enzyme co-localization or metabolon formation and would directly inform the concern raised above. Even partial overlap in localization (e.g., cytosolic versus vacuolar interfaces) would provide valuable context for understanding pathway organization *in vivo*. Alternatively, the results may provide evidence that such a complex does not form. In either case, localization data would substantially strengthen the interpretation of pathway organization and enzyme cooperation *in vivo*.

3. For the enzyme assays using deuterium-labeled substrates, did the authors observe any kinetic or product-distribution differences between labeled and unlabeled substrates that would indicate an isotope effect? If present, such effects could provide additional mechanistic insight into rate-limiting steps or hydride transfer chemistry in the pathway, and merit explicit discussion.

4. In the sentence beginning “Labelled precursor feeding studies indicate that that 2 undergoes C6-reduction...”, there is a duplicated “that.” One instance should be removed.

Reviewer #2

(Remarks to the Author)

In this manuscript from Schwabe and colleagues, the complete nicotine biosynthesis pathway for *Nicotiana tabacum* is elucidated by identifying glycosylation of key pathway substrates, followed by deglycosylation in later steps. They combine in vitro experiments with in planta experiments to confirm the reactions and their substrates and report X-ray crystal structures of the two oxidoreductases (A622 and BBLa) bound to pathway intermediates. It is a commendably thorough manuscript. This research fills a longstanding gap in the field, and the findings are exciting and likely to be of interest to a broad audience. My suggestions for improving the manuscript are as follows:

1. The introduction mentions both BBLa and b, but only a is characterized in the manuscript. Could the authors expand on why only BBLa was selected for characterization within the nicotine biosynthesis pathway?
2. I recommend that the authors submit the UGT sequence to the UGT Nomenclature Committee to receive standardized name for UGT1. This is a very large protein family, so having an assigned nomenclature may be helpful for distinguishing this protein from others in *N. tabacum*.
3. UGTs can accept a wide variety of substrates, but the manuscript does not include information for substrates tested (i.e., other alkaloids, terpenes). It is interesting that the UGT glycosylates the amine of nicotinic acid, but maybe it has broad activity with amines and other functional groups (hydroxyl groups, etc.). Since it is also expressed in tissues where nicotine is not produced, it may be likely that it can catalyze many reactions.
4. Similarly, was the sugar donor specificity of UGT1 tested using co-substrates besides UDP-glucose?
5. The discussion would benefit from providing more context for the novelty of the finding by providing other instances where cryptic glycosylation/deglycosylation is found in pathways.
6. The methods should include information about how protein concentration was determined after protein purification.
7. For the protein crystallization, please include the concentration of protein that was used, droplet volume, and the ratio of protein:crystallization condition in each droplet.
8. The last two sentences of "Metabolite analysis of *N. benthamiana* pathway reconstruction" are redundant.
9. Line 185– Percent similarity for NicGS to the reticuline oxidase should be reported.

Reviewer #3

(Remarks to the Author)

The paper by Schwabe et al. investigates the biosynthesis of nicotine, a well-known alkaloid whose pathway has remained incomplete. The authors provide compelling characterization of two newly identified enzymes: a glycosyltransferase and a β -glucosidase, along with two previously partially characterized enzymes, a NADPH-dependent reductase and a BBE-like oxidase. Although earlier work had suggested the presence of a glycosylation step, this study represents the first complete and convincing elucidation of the nicotine biosynthetic pathway. I view this as a major discovery in plant specialized metabolism, supported by well-designed and carefully executed experiments.

The in vitro reconstitution clearly demonstrates pathway function. The stereochemistry is shown to be determined by the BBE-like oxidase, consistent with both the data presented here and previous knockout studies. Importantly, the involvement of cryptic glycosylation and subsequent deglycosylation is supported by both biochemical assays and structural insights from the reductase and the BBE-like oxidase, who take them as substrates. Although the in vivo reconstruction in tobacco is influenced by endogenous glycosyltransferase activity, this does not, in my view, undermine the overall design or conclusions.

I have only minor suggestions and editorial comments:

1. Clarity and readability. The manuscript is written in a very concise style. Given that Nature Communications allows ample space for text and figures, I encourage the authors to expand certain sections to improve readability. Several Extended Data figures, e.g. Extended Data Fig. 7, could be incorporated into the main figures to reduce back-and-forth navigation.
2. Are mutants available for the glycosyltransferase or β -glucosidase? If so, brief analysis would further strengthen the study. While not essential for the main conclusions, such data would add valuable in vivo validation.
3. Instability and reactivity of intermediates. The authors may elaborate on the instability of compound 4b/5b, the mechanism underlying its spontaneous oxidation to 12. I assume targeted searches for masses corresponding to 4b/5b were performed, but this is not fully clear from the current text. Is 4b/5b ever detected, even in small quantity? Also the author may elaborate

on the mechanism of decarboxylation after reduction. These are the unclear parts of the pathway, I encourage the author to elaborate on these and provide more insights.

4. The manuscript concludes that the BBE-like oxidase binds and catalyzes oxidation of both 4b and 5b. It is not clear to me with current condensed writing style.

5. The discussion could be extended by discussing some of the above points and enriched by referencing other recent discoveries involving unexpected glycosylated or protected intermediates, such as in steroidal alkaloid biosynthesis, therefore placing this work in a broader metabolic context.

Overall, this is an excellent and impactful study that significantly advances our understanding of nicotine biosynthesis. With minor clarifications and expanded discussion, it will be more accessible and compelling to the community.

Version 1:

Reviewer comments:

Reviewer #1

(Remarks to the Author)

I thank the authors for their thorough and transparent responses to my comments. The revisions are mostly satisfactory and I recommend publication without further delay. For my comment 2 on subcellular localization, the authors attempted mCherry fusion experiments for all four enzymes but encountered technical challenges across the board, apparent cleavage artefacts for NaGT and A622 fusions, trafficking overload for BBL, and absence of signal for β -GD—and accordingly no new localization data appear in the manuscript. In lieu of this, they have expanded the textual discussion of existing evidence (cytosolic A622, vacuolar BBL), added prediction-based arguments for NicGH vacuolar targeting, and incorporated the Liu et al. 2026 β -glucosidase data with appropriate caveats. I appreciate the candor with which these experimental difficulties are disclosed, and I accept that localization of this pathway is more suited for future work. The gap is real but does not, in my view, detract from the central conclusions of the paper.

Reviewer #2

(Remarks to the Author)

The authors have sufficiently incorporated my comments into the manuscript, and I have no further comments.

Reviewer #3

(Remarks to the Author)

The updated manuscript has sufficiently addressed my previous comments. I congratulate the authors on their interesting findings.

Response to Referees

Reviewer #1 (Remarks to the Author):

This paper by Schwabe et al. resolves a longstanding mystery in plant specialized metabolism by elucidating the unknown terminal steps of biosynthetic pathway to nicotine in *Nicotiana*. Although “nicotine synthase” activity had been demonstrated decades ago using crude enzyme preparations, the molecular identity and mechanistic basis of this activity remained unknown. In this paper, the authors reconstitute nicotine biosynthesis *in vitro* and *in planta* using four defined enzymes and, also provide high-resolution crystal structures of two key catalytic components that illuminate the underlying enzymatic mechanism at the atomic level. Together, the biochemical reconstitution, isotope-labeling experiments, structural biology, and plant validation represent a novel and significant advance in plant natural product biochemistry. The work is carefully designed and well executed, and the findings have clear translational implications for metabolic engineering, synthetic biology, and biocatalysis of pyridine alkaloids.

We thank the reviewer for such a positive assessment of this manuscript.

1. My main comment is the use of the term “nicotine synthase.” In the historical literature, this term originated from crude biochemical activity assays, without knowledge of the molecular nature of the catalyst. Conventionally, “synthase” implies either a single enzyme or a defined enzyme complex. In the present study, four enzymes are collectively described as nicotine synthase, but it remains unclear whether these proteins form a stable complex, a transient metabolon, or act entirely independently.

The authors could address this point experimentally by testing whether the recombinant enzymes associate with one another, for example via co-expression followed by size-exclusion chromatography, native PAGE, or pull-down assays. Demonstrating a stable or semi-stable complex would strengthen the rationale for using the term “nicotine synthase,” particularly given the instability of proposed intermediates and the potential benefit of substrate channeling to enhance catalytic efficiency and suppress side reactions. If no physical association exists, I suggest the authors refrain from referring to the four enzymes collectively as “nicotine synthase,” and instead describe them as a multi-enzyme biosynthetic cascade/pathway.

This is a reasonable and thoughtful point. Our main intention using the term “nicotine synthase” was to link our work directly to the Friesen and Leete 1990 report of a crude enzyme preparation that they named “nicotine synthase” which was able to form nicotine from nicotinic acid and N-methylpyrrolinium (Brent Friesen and Leete 1990). This is exactly what our enzymatic cascade is capable of doing and so to demonstrate we “solved” this question we used their nomenclature. We never intended to suggest that these enzymes were operating as a stable complex or metabolon but understand that in modern terminology this is the impression that was given. We have consequently decided to not use the term “nicotine synthase” to describe the combined action of the four enzymes (NaGT, NaGR, NicGS, NicGH) and instead refer to them as the “nicotine biosynthesis cascade” or “nicotine cascade”.

The question of protein interactions remains interesting especially in the context of the unstable intermediates. However, it is near impossible that A622 and BBL can physically interact given their known subcellular compartmentalisation: A622 is cytoplasmic (Shoji et al. 2002) and BBL is vacuolar (Kajikawa et al. 2011). Furthermore the “nicotine synthase” complex reported by Friesen and Leete underwent very minimal purification that is insufficient to suggest a defined enzyme complex (Brent Friesen and Leete 1990). We have added a section in the discussion which discusses the nature of “nicotine synthase” in the context of subcellular localisation and the Friesen and Leete experiments (line 360).

The division of the nicotine biosynthetic cascade across subcellular compartments—the cytosol and the vacuole—does not support the hypothesis that the cascade can operate as a structurally defined multi-enzyme complex. Furthermore, the reported “nicotine synthase” enzyme isolated from tobacco roots by Friesen and Leete in 1990 was obtained with minimal purification: just filtration, desalting and crude ammonium sulfate protein fractionation, with activity observed in the 25-75% fraction¹⁶. Therefore, whilst the nicotine biosynthetic cascade enzymes (NaGT, NaGR, NicGS, NicGH) and their cofactors must have been present and soluble in this broad fraction, without further purification or alternative evidence there is no reason to presume that the cascade acts as a “nicotine synthase” enzyme complex.

We have not ruled out a complex organisation of the pathway in plant cells, and we are working with collaborators to further characterise the pathway in planta, including understanding the connection between subcellular compartmentalisation and unstable intermediates. However, this is a complex study still at early stages and using novel techniques and is unsuitable to be included in this manuscript.

2. I encourage the authors to determine and present the subcellular localization of the enzymes in *Nicotiana*. Such data would help corroborate the possibility of enzyme co-localization or metabolon formation and would directly inform the concern raised above. Even partial overlap in localization (e.g., cytosolic versus vacuolar interfaces) would provide valuable context for understanding pathway organization in vivo. Alternatively, the results may provide evidence that such a complex does not form. In either case, localization data would substantially strengthen the interpretation of pathway organization and enzyme cooperation in vivo.

*We completely agree with the importance of subcellular localisation, and have attempted to characterise this. To do this we generated C-terminal mCherry fusions of all four enzymes from *N. tabacum* (NaGT, NaGR, NicGS, NicGH) and transiently expressed these in the leaves of *N. benthamiana* and imaged them 3 days post-infiltration by confocal microscopy. We also co-infiltrated CFP markers for either the ER or tonoplast to enable identification of subcellular localisation. Unfortunately, we ran into a number of issues with this experiment which could not be easily rectified and render the results inadequate for publication:*

- *The UGT and A622 constructs showed fluorescence in the cytosol as expected, but also a strong nuclear fluorescence signal. The size of our fluorescent fusion would be expected to exceed the cutoff of diffusion into the nucleus. This strongly suggests cleavage of mCherry from our fusion protein, rendering these results unreliable.*
- *The BBL-mCherry construct showed strong fluorescence in both the ER and cytosol, suggesting that the transient overexpression overloaded the cell's trafficking*

machinery. The C-terminal fusion also obscured a short predicted C-terminal vacuolar sorting signal which is likely important for correct trafficking to the vacuole.

- The bGD-mCherry unfortunately showed no fluorescence at all. Possibly due to misfolding and protein degradation.

We intend to explore the subcellular localisation/compartmentalisation of the pathway further. However, given the experimental challenges we have faced this will not be straightforward and should instead form part of a larger future story into in planta characterisation of the pathway. However, we have added more discussion/emphasis to the manuscript of what is already known of the subcellular localisation. In particular, the localisation of A622 and BBL are known to be cytosolic and vacuolar respectively. We have added to the introduction to state that A622 is “a cytosolically localised isoflavone reductase-like enzyme³²” (line 60) and comment on this in the discussion (paragraph starting “As well as modulating ...” (line 347).

As well as modulating reactivity, the detoxification effect of glucoside-conjugates in plants can be attributed to driving vacuolar localisation of metabolites. This may also be true for nicotine biosynthesis, given the known subcellular localisation of NaGR (A622) to the cytosol³², and NicGS (BBL) to the vacuole³⁰, and also given that nicotine accumulates in vacuoles of tobacco root cells during biosynthesis³⁸.

A recent paper, released during the revision of this manuscript, reported the identification of three methyl jasmonate responsive beta-glucosidases from tobacco (Liu *et al.* 2026). These three beta-glucosidases are the three homologs of the enzyme we have characterised as NicGH, the final step in nicotine biosynthesis. We describe this paper in the discussion from line 300.

During the revision of this manuscript, a report was published describing the systematic assessment of β -glucosidase genes in *N. tabacum*⁵¹. It revealed that the expression of NicGH (β -GD1) and its paralogs β -GD2 and β -GD3 (named NtBGLU51, 50 and 49 respectively in the β -GD manuscript) were all strongly induced by methyl jasmonate, supporting their role in defence response.

The authors report that one of these homologs (NtBGLU50 / our β -GD2) is ER-localised in *N. benthamiana* transient expression experiments despite having a vacuolar signalling signal. We believe that this is an artefact due to misprocessing of their fusion protein in the ER: for example, our DeepLoc2.0 prediction suggests that as well as the long N-terminal signal peptide, there is also a C-terminal vacuolar sorting signal, which may be important for correct subcellular localisation (Fig S1). This is obscured by their C-terminal fluorescent protein fusion. We have added discussion of this specific data in our manuscript and highlight the need for further investigations into subcellular localisation of these enzymes (see line 351).

The predicted N-terminal signal peptide and C-terminal vacuolar sorting signal of NicGH strongly suggest it will also localise to the vacuole (Fig. S1). The recent manuscript reporting tobacco β -glucosidases included an experiment where GFP was fused to the C-terminus of β -GD2 (NtBGLU50) and the fusion protein was transiently expressed in *N. benthamiana*.

Surprisingly, the protein appeared to localise to the endoplasmic reticulum, despite its predicted vacuolar localisation sequence. Further investigation of subcellular localisation of β -GDs is required for a conclusive outcome.

3. For the enzyme assays using deuterium-labeled substrates, did the authors observe any kinetic or product-distribution differences between labeled and unlabeled substrates that would indicate an isotope effect? If present, such effects could provide additional mechanistic insight into rate-limiting steps or hydride transfer chemistry in the pathway, and merit explicit discussion.

Our current experiments have focused on end-point product distributions rather than kinetics of the cascade. We see evidence for the kinetic isotope effect (KIE) in the difference in the d3/d4 ratio between the assays with and without the NicGS (BBLa) enzyme. In particular d4 predominates in the absence of the NicGS enzyme due to the KIE. In the presence of the NicGS, d3-isotopologues predominate due to the diastereoselective removal of the hydride by this enzyme. This labelling is consistent with NaGR (A622) catalyzed reduction of nicotinic acid glucoside (10) and the NicGS (BBLa) catalyzed oxidation of 1,2-dihydronicotine glucoside (5b) both occurring at C6, with a protide introduced by A622 and the deuteride abstracted by BBLa. In contrast, non-enzymatic oxidation favors protide abstraction due to the kinetic isotope effect, as seen in the abundance of more highly deuterated products (e.g. 1-d4 and 1-d6 from 2-d4 and 2-1d6 respectively) in the absence of NicGS (BBLa) (Fig. 5A-C, Table S6), and in the predominance of 12-d4 in the 2-d4 fed pathway (Fig. 5). We have added more detail describing these results around lines 170-205.

Feeding experiments with isotopically labelled precursors have been crucial in establishing atomic level information about nicotine biosynthesis, such as the loss of the C6 hydrogen from nicotinic acid (Fig. 1)^{17-19,31}. We reconstructed the four-enzyme nicotine biosynthesis cascade using isotopically labelled nicotinic acid-(ring-d4) (2-d4) as a substrate (Fig. 5, Fig. S8, Table S6). With the full cascade, we observed accumulation of the major product 1-d3 (Fig. 5B, row i). A similar labelling pattern is observed for nicotine glucoside (13) in the absence of NicGH (Fig. 5B, row ii). In contrast, when NicGS (BBLa) was absent then not only is the yield of 1 reduced, but the labelling pattern of 1 is changed with 1-d4 now the major isotopologue (Fig. 5B, row iii). This suggests that NicGS is responsible for selective formation of 1-d3 in the full cascade.

To pinpoint exactly which deuterium was being exchanged, we used the singly labelled nicotinic acid-d6 (2-d6). With NicGS present, we saw formation of the major product 1-d0; without NicGS, 1-d1 was the major product (Fig. 5C). This verifies that the isotope exchange determined by NicGS occurs at position C6 (Fig. 5C). This labelling pattern, where the C6 deuterium from labelled 2 is lost in the product 1, is consistent with a cascade where NaGR (A622) catalyses reduction of 2, transferring an unlabelled hydride (i.e. a protide) from NADPH onto C6, followed by the NicGS (BBLa) catalysing specific removal of the deuteride from the intermediate 1,2-dihydronicotine glucoside (5b) (Fig. 5A). The overall effect is a swapping of the deuteride for protide at C6, consistent with previous observations^{17-19,31}.

In the absence of NicGS, the ratio of isotopologues is not equal, but reversed, with the dominant isotopologue products being the fully labelled 1-d1 or 1-d4 from 2-d6 or 2-d4 respectively (Fig. 5B, row iii, and Fig. 5C). This indicates that the non-enzymatic oxidation of 5b favours the abstraction of the unlabelled protide (introduced by NaGR from NADPH) rather than the deuteride (from the labelled substrate 2). This bias occurs due to the kinetic isotope effect which renders carbon-

deuterium bonds stronger than carbon-protium bonds, so the latter are broken more readily in the non-enzymatic reaction where selectivity is determined by bond energy rather than spatial context.

When **2-d₄** is used a substrate, the NicGS (BBLa) also appears to have an effect on the labelling of pyridine glucoside **12**, which accumulates in the absence of *N*-methylpyrrolinium (**3**). Without NicGS (BBLa), the ratio of fully labelled **12-d₄** to partially labelled **12-d₃** is significantly greater than when NicGS (BBLa) is present (Fig. 5B, rows *iv* and *v*, Table S6). This implies that NicGS can bind and catalyse the oxidation **4b** as well as **5b**. Cascade formation of nornicotine-d₃ (**15-d₃**) and anabasine-d₃ (**6-d₃**) from **2-d₄** confirmed they are also formed via NicGS (BBLa)-catalysed oxidation (Fig. 5D and E).

It is likely that the deuterated substrate does slow the NicGS enzyme compared with the unlabelled substrate. However, NicGS is not the rate limiting step of the cascade (likely this is the reduction of nicotinic acid N-glucoside by NaGR). Due to the nature of the cascade we cannot assay the NicGS activity in the absence of NaGR (A622), so cannot observe the KIE on the rate of NicGS catalysed oxidation.

4. In the sentence beginning "Labelled precursor feeding studies indicate that that **2** undergoes C6-reduction...", there is a duplicated "that." One instance should be removed.

Thank you - we have corrected this.

Reviewer #2 (Remarks to the Author):

In this manuscript from Schwabe and colleagues, the complete nicotine biosynthesis pathway for *Nicotiana tabacum* is elucidated by identifying glycosylation of key pathway substrates, followed by deglycosylation in later steps. They combine *in vitro* experiments with *in planta* experiments to confirm the reactions and their substrates and report X-ray crystal structures of the two oxidoreductases (A622 and BBLa) bound to pathway intermediates. It is a commendably thorough manuscript. This research fills a longstanding gap in the field, and the findings are exciting and likely to be of interest to a broad audience.

We thank the reviewer for their positive comments.

My suggestions for improving the manuscript are as follows:

1. The introduction mentions both BBLa and b, but only a is characterized in the manuscript. Could the authors expand on why only BBLa was selected for characterization within the nicotine biosynthesis pathway?

*N. tabacum is derived from a hybridization event between *N. sylvestris* and *N. tomentosiformis*, giving it an allotetraploid genome. This means that all of the nicotine biosynthesis genes have at least one homolog, from each of *N. sylvestris* and *N. tomentosiformis*, such as A622 and A622L. Likewise, BBLa and BBLb have 93% protein sequence similarity and share this homeologous origin - BBLa from *N. sylvestris* and BBLb from *N. tomentosiformis*. The same is true for the newly described bGDs: bGD1 and bGD2 are homeologs, and bGD3 is a close paralog.*

*Previous genome editing work in *N. tabacum* has shown the genetic equivalence of A622/A622L and BBLa/BBLb, as to guarantee low nicotine lines through genome editing then both homeologs need to be knocked-out. Furthermore, the genes show the same expression patterns and near identical sequences, providing evidence they are functionally equivalent. For expediency, given the many highly similar homologs and our *in vitro* approach, we elected to test a characteristic homolog for each enzyme in our *in vitro* work.*

This comment has led us to more openly discuss our use of single representative paralogs for A622, BBL and bGD. We have added a comment to line 109 stating: "For expediency, we elected to test a single representative paralog of each enzyme type." We have also added a paragraph in the discussion from line 291 which justifies the use of single representative paralogs by describing the close sequence identities, similar expression patterns and genetic evidence.

*Whilst we elected to test only a single representative paralog for each enzyme type in our *in vitro* assays, there is considerable evidence to suggest that the highly similar paralogs (or homeologs, i.e. paralogs derived from whole genome duplications) are functionally equivalent. NaGR (A622) has 95.5% protein sequence identity to its homeolog A622L and has similar expression patterns (Table S2). Genome editing experiments show that both homeologs must be knocked-out to guarantee ultra-low nicotine content in mutant tobacco lines^{28,50}. Similarly, BBLa (NicGS) and BBLb are homeologs with 92.8% protein sequence identity, similar expression patterns (Fig. 2C) and genome editing experiments implicate both genes in the formation of (S)-nicotine³⁴. The*

newly described NicGH (β -GD1) shares 91.8% with its homeolog β -GD2 and 83.6% protein sequence identity with its non-syntenic paralog β -GD3; all genes have similar expression patterns across tissues (Fig. 2C). During the revision of this manuscript, a report was published describing the systematic assessment of β -glucosidase genes in *N. tabacum*⁵¹. It revealed that the expression of NicGH (β -GD1) and its paralogs β -GD2 and β -GD3 (named NtBGLU51, 50 and 49 respectively in the β -GD manuscript) were all strongly induced by methyl jasmonate, supporting their role in defence response.

Furthermore, please note that we are currently performing a more in-depth assessment of BBLs paralogs from N. tabacum and will be testing BBLb as part of this work, but this work is ongoing and is beyond the scope of this manuscript.

2. I recommend that the authors submit the UGT sequence to the UGT Nomenclature Committee to receive standardized name for UGT1. This is a very large protein family, so having an assigned nomenclature may be helpful for distinguishing this protein from others in *N. tabacum*.

Thank you for this suggestion. We have submitted the NicGT (UGT1) sequence to the nomenclature committee and received a new name: UGT709L18. This is now included in the manuscript, described in line 103.

3. UGTs can accept a wide variety of substrates, but the manuscript does not include information for substrates tested (i.e., other alkaloids, terpenes). It is interesting that the UGT glycosylates the amine of nicotinic acid, but maybe it has broad activity with amines and other functional groups (hydroxyl groups, etc.). Since it is also expressed in tissues where nicotine is not produced, it may be likely that it can catalyze many reactions.

This is a good point. UGTs are known to be promiscuous and are indeed implicated in detoxification and movement into the vacuole of metabolites. It is plausible that this is related to the evolution of the nicotine biosynthetic pathway, with duplication of upstream genes leading to overaccumulation of nicotinic acid and resulting in glucosylation for detoxification. We have already tested some other plausible substrates (Figure S4) showing activity on pyridine, to give pyridine N-glucoside, but no activity on nicotine. We have improved our description of this in the manuscript at line 119.

NaGT could also catalyse the formation of pyridine glucoside (**12**) from pyridine and UDP-glucose; but could not form nicotine glucoside (**13**) from nicotine and UDP-glucose (Fig. S4).

Testing other substrates with the UGT that are not clearly connected to nicotine biosynthesis is outside the scope of this nicotine-focussed manuscript, but would be an interesting question for future work if we were to attempt to expand the cascade further.

4. Similarly, was the sugar donor specificity of UGT1 tested using co-substrates besides UDP-glucose?

We did not test the UGT NaGT with UDP-sugar donors aside from UDP-glucose. Given the accumulation of nicotinic acid N-glucoside in A622 knock out plants, the glucoside is clearly

the relevant glycoside in planta (Kajikawa, Hirai, and Hashimoto 2009). We also observe both pyridine and nicotine glucosides in the plant, further indicating this is the natural glycosyl adduct (Fig. 8). Characterisation of the NaGT (UGT709L18) with substrates not directly connected to nicotine biosynthesis is beyond the scope of this manuscript.

5. The discussion would benefit from providing more context for the novelty of the finding by providing other instances where cryptic glycosylation/deglycosylation is found in pathways.

This is a valid request, and we are grateful to have the opportunity to describe our work's novelty in more detail. We have added three new paragraphs that discuss cryptic glycosylation in the context of natural product biosynthesis, from 311 to 359.

Glucosylation steps are common in natural product biosynthesis, but the use of the glucoside in nicotine biosynthesis as an activating group is distinct from other known examples of glucosylation. Natural product glucose conjugates are usually seen in the context of detoxification, allowing the host plant to store otherwise highly toxic molecules as inactivated glucosides, ready for future activation by deglycosylation⁵². For example, the benzoxazinoids DIBOA and DIMBOA are produced as defence compounds in grasses. These are sequestered in the plant vacuole as inactivated glucosides, which can be released by a specific glycosyl hydrolase for defence related purposes⁵³. This kind of conditional detoxification/activation is also seen for cyanogenic glucosides e.g. in sorghum⁵⁴ and in the regulation of brassinosteroid and cytokinin hormone homeostasis⁵⁵.

In plant metabolism, glucose esters are commonly used as acyl donors. This involves a glucose moiety being added to a carboxylic acid, and the glucose ester product undergoes transesterification, a step catalysed by serine carboxypeptidase-like acyltransferases (SCPL-ATs)⁵⁶. This is analogous to coenzyme A thioesters, which are acyl donors that undergo transesterification catalysed by BAHD-acyltransferases⁵⁶. Glucose ester intermediates feature in the biosynthesis of sinapate (sinapoylcholine) and the UV-protectant sinapoylmalate from sinapic acid via the activated sinapoylglucose ester intermediate in Brassicaceae⁵⁷, as well as in hormone homeostasis forming IAA-myo-inositol conjugates via indoleacetic acid-glucoside⁵⁸. Interestingly, a glucoside ester is an intermediate in tropane alkaloid biosynthesis, a pathway from Solanaceae species that shares many features with nicotine biosynthesis. In particular, a UGT glucosylates phenyllactate to phenyllactylglucoside, enabling SCPL-AT catalysed transesterification to littorine with tropine⁵⁹. The glucosyl group in glucose esters acts to activate the carboxylic acid by forming an ester and becoming a favourable leaving group, with glucose being a product of the transesterification reaction. This is distinct from the proposed glucose activation in nicotine biosynthesis where the glucose is not a direct product of the core reaction but instead modulates the activity of intermediates and requires a dedicated removal step: NicGH catalysed hydrolysis.

A cryptic glycosylation was recently described in Solanaceae steroidal glycoalkaloid biosynthesis, involving the cholesterol glucuronosyl transferase GAME15^{60,61}. A cryptic glucosylation was also discovered in the biosynthesis of the yeast antibiotic anisomycin⁶². In both of these examples it appears that the glycosyl moiety acts mostly as a protecting group/recognition handle for the downstream biosynthetic enzymes, rather than having an activation effect as we propose for nicotine biosynthesis.

As well as modulating reactivity, the detoxification effect of glucoside-conjugates in plants can be attributed to driving vacuolar localisation of metabolites. This may also be true for nicotine biosynthesis, given the known subcellular localisation of NaGR (A622) to the cytosol³², and NicGS (BBL) to the vacuole³⁰, and also given that nicotine accumulates in vacuoles of tobacco

root cells during biosynthesis³⁸. The predicted *N*-terminal signal peptide and *C*-terminal vacuolar sorting signal of NicGH strongly suggest it will also localise to the vacuole (Fig. S1). The recent manuscript reporting tobacco β -glucosidases included an experiment where GFP was fused to the *C*-terminus of β -GD2 (NtBGLU50) and the fusion protein was transiently expressed in *N. benthamiana*. Surprisingly, the protein appeared to localise to the endoplasmic reticulum, despite its predicted vacuolar localisation sequence. Further investigation of subcellular localisation of β -GDs is required for a conclusive outcome.

6. The methods should include information about how protein concentration was determined after protein purification.

Thank you - this has now been added around line 440: "Protein concentration was estimated using a Nanodrop spectrophotometer with predicted extinction co-efficients derived from ExPASy ProtParam, and then stored at -70 °C."

7. For the protein crystallization, please include the concentration of protein that was used, droplet volume, and the ratio of protein:crystallization condition in each droplet.

Information about protein concentration for crystallisation is now included around line 500 for NaGR, "Drops (300 nL) were set up with a 1:1 mixture of 12 mg/mL NaGR (A622) with precipitant mother liquor." and 509 for NicGS "After incubation, the deglycosylated BBL was used for crystallisation at 15 mg/mL".

8. The last two sentences of "Metabolite analysis of *N. benthamiana* pathway reconstruction" are redundant.

Thank you - the second sentence is now removed.

9. Line 185-- Percent similarity for NicGS to the reticuline oxidase should be reported.

We have now added this; they share 44.6% protein sequence identity (line 229).

Reviewer #3 (Remarks to the Author):

The paper by Schwabe *et al.* investigates the biosynthesis of nicotine, a well-known alkaloid whose pathway has remained incomplete. The authors provide compelling characterization of two newly identified enzymes: a glycosyltransferase and a β -glucosidase, along with two previously partially characterized enzymes, a NADPH-dependent reductase and a BBE-like oxidase. Although earlier work had suggested the presence of a glycosylation step, this study represents the first complete and convincing elucidation of the nicotine biosynthetic pathway. I view this as a major discovery in plant specialized metabolism, supported by well-designed and carefully executed experiments.

The *in vitro* reconstitution clearly demonstrates pathway function. The stereochemistry is shown to be determined by the BBE-like oxidase, consistent with both the data presented here and previous knockout studies. Importantly, the involvement of cryptic glycosylation and subsequent deglycosylation is supported by both biochemical assays and structural insights from the reductase and the BBE-like oxidase, who take them as substrates. Although the *in vivo* reconstruction in tobacco is influenced by endogenous glycosyltransferase activity, this does not, in my view, undermine the overall design or conclusions.

I have only minor suggestions and editorial comments:

1. Clarity and readability. The manuscript is written in a very concise style. Given that Nature Communications allows ample space for text and figures, I encourage the authors to expand certain sections to improve readability. Several Extended Data figures, e.g. Extended Data Fig. 7, could be incorporated into the main figures to reduce back-and-forth navigation.

To improve readability, we have added new main text figures, including Extended Data Fig. 7 as the new Fig. 9, (adapted from previous Extended Data figures). The remaining extended data figures and tables have been moved to the supplementary information. We have also added a considerable amount of text to the discussion section and have expanded the description of the results as requested below.

2. Are mutants available for the glycosyltransferase or β -glucosidase? If so, brief analysis would further strengthen the study. While not essential for the main conclusions, such data would add valuable *in vivo* validation.

*Unfortunately, no mutants are available for the NaGT glycosyltransferase or the NicGH β -glucosidase at this point. We are working to generate these in *N. benthamiana* and also attempting root VIGS to silence these genes, but anticipate a long timescale. We expect the results from these to be part of a future body of work that analyses this pathway in the plant context. We acknowledge that such *in vivo* information would be valuable for *in planta* validation but as the reviewer notes, their inclusion is not essential for the main conclusions of the paper.*

3. Instability and reactivity of intermediates. The authors may elaborate on the instability of compound 4b/5b, the mechanism underlying its spontaneous oxidation to 12. I assume targeted searches for masses corresponding to 4b/5b were performed, but this is not fully clear from the current text. Is 4b/5b ever detected, even in small quantity? Also the author

may elaborate on the mechanism of decarboxylation after reduction. These are the unclear parts of the pathway, I encourage the author to elaborate on these and provide more insights.

We have expanded the text and figures to elaborate on these points more clearly. We note that we checked for the masses but could not observe peaks corresponding to the unstable intermediate 4b (line 124).

No peak corresponding to the expected mass of 1,2-dihydropyridine glucoside (**4b**) ($[M+H]^+ = m/z$ 244) could be observed in NaGR catalysed reactions.

We did see a peak of the mass for 5b but this was revealed to be dihydrometanicotine N-glucoside (Glc-11). We describe this in line 136.

We were also able to identify a small peak matching the expected mass of dihydronicotine glucoside (**5b**) ($[M+H]^+ = m/z$ 327) but this peak was assigned as its isomer dihydrometanicotine N-glucoside (Glc-11) via comparison to a chemically synthesised standard (Fig. S5).

We have added a description of how the decarboxylation mechanism is likely to occur (line 126-129) and added a figure panel (Fig S5C) showing this.

The formation of **4b** from dihydronicotinic acid N-glucoside (**14**) could occur non-enzymatically through an enamine-imine tautomerisation followed by decarboxylation (Fig. S5C).

We also added that oxidation likely occurs through reaction with molecular oxygen (line 129).

The subsequent oxidation of **4b** to form **12** likely occurs through reaction with molecular oxygen.

In addition, we have added a figure panel showing formation of dihydrometanicotine at the relevant part of the text (line 140, Fig S5D).

We propose that (*R,S*)-dihydronicotine glucoside (**5b**) can either decompose into Glc-11 or oxidise into (*R,S*)-**13**. The former can occur through a series of tautomerisation steps forming Glc-11 and be followed by hydrolysis to yield **11** (Fig. S5D).

4. The manuscript concludes that the BBE-like oxidase binds and catalyzes oxidation of both 4b and 5b. It is not clear to me with the current condensed writing style.

Indeed the NicGS appears to catalyse the oxidation of both 4b and 5b. This was determined through the examination of the isotope labelling experiments and the ratio of the isotope products. To make this more clear we have rewritten the whole isotope labelling section (line 170-205) and improved the presentation of the data (Fig 5), clearly describing in the text which sections of the figure correspond to the statements.

Feeding experiments with isotopically labelled precursors have been crucial in establishing atomic level information about nicotine biosynthesis, such as the loss of the C6 hydrogen from nicotinic acid (Fig. 1)^{17–19,31}. We reconstructed the four-enzyme nicotine biosynthesis cascade using isotopically labelled nicotinic acid-(ring-d₄) (2-d₄) as a substrate (Fig. 5, Fig. S8, Table S6). With the full cascade, we observed accumulation of the major product 1-d₃ (Fig. 5B, row *i*). A similar labelling pattern is observed for nicotine glucoside (13) in the absence of NicGH (Fig. 5B, row *ii*). In contrast, when NicGS (BBLa) was absent then not only is the yield of 1 reduced, but the labelling pattern of 1 is changed with 1-d₄ now the major isotopologue (Fig. 5B, row *iii*). This suggests that NicGS is responsible for selective formation of 1-d₃ in the full cascade.

To pinpoint exactly which deuterium was being exchanged, we used the singly labelled nicotinic acid-d₆ (2-d₆). With NicGS present, we saw formation of the major product 1-d₀; without NicGS, 1-d₁ was the major product (Fig. 5C). This verifies that the isotope exchange determined by NicGS occurs at position C6 (Fig. 5C). This labelling pattern, where the C6 deuterium from labelled 2 is lost in the product 1, is consistent with a cascade where NaGR (A622) catalyses reduction of 2, transferring an unlabelled hydride (i.e. a protide) from NADPH onto C6, followed by the NicGS (BBLa) catalysing specific removal of the deuteride from the intermediate 1,2-dihydronicotine glucoside (5b) (Fig. 5A). The overall effect is a swapping of the deuteride for protide at C6, consistent with previous observations^{17–19,31}.

In the absence of NicGS, the ratio of isotopologues is not equal, but reversed, with the dominant isotopologue products being the fully labelled 1-d₁ or 1-d₄ from 2-d₆ or 2-d₄ respectively (Fig. 5B, row *iii*, and Fig. 5C). This indicates that the non-enzymatic oxidation of 5b favours the abstraction of the unlabelled protide (introduced by NaGR from NADPH) rather than the deuteride (from the labelled substrate 2). This bias occurs due to the kinetic isotope effect which renders carbon-deuterium bonds stronger than carbon-protium bonds, so the latter are broken more readily in the non-enzymatic reaction where selectivity is determined by bond energy rather than spatial context.

When **2-d₄** is used a substrate, the NicGS (BBLa) also appears to have an effect on the labelling of pyridine glucoside **12**, which accumulates in the absence of *N*-methylpyrrolinium (**3**). Without NicGS (BBLa), the ratio of fully labelled **12-d₄** to partially labelled **12-d₃** is significantly greater than when NicGS (BBLa) is present (Fig. 5B, rows *iv* and *v*, Table S6). This implies that NicGS can bind and catalyse the oxidation **4b** as well as **5b**. Cascade formation of nornicotine-d₃ (**15-d₃**) and anabasine-d₃ (**6-d₃**) from **2-d₄** confirmed they are also formed via NicGS (BBLa)-catalysed oxidation (Fig. 5D and E).

5. The discussion could be extended by discussing some of the above points and enriched by referencing other recent discoveries involving unexpected glycosylated or protected intermediates, such as in steroidal alkaloid biosynthesis, therefore placing this work in a broader metabolic context.

We have improved the clarity of the results section, in particular the sections on the in vitro reconstitution of the cascade and the isotope labelling. We have added extra figure panels showing the non-enzymatic mechanisms (Fig S5C and D). We have also added considerable extra detail to the discussion section on glucosylation, including references to steroidal alkaloid biosynthesis (specifically lines 340-345).

Glucosylation steps are common in natural product biosynthesis, but the use of the glucoside in nicotine biosynthesis as an activating group is distinct from other known examples of glucosylation. Natural product glucose conjugates are usually seen in the context of detoxification, allowing the host plant to store otherwise highly toxic molecules as inactivated glucosides, ready for future activation by deglycosylation⁵². For example, the benzoxazinoids DIBOA and DIMBOA are produced as defence compounds in grasses. These are sequestered in the plant vacuole as inactivated glucosides, which can be released by a specific glycosyl hydrolase for defence related purposes⁵³. This kind of conditional detoxification/activation is also seen for cyanogenic glucosides e.g. in sorghum⁵⁴ and in the regulation of brassinosteroid and cytokinin hormone homeostasis⁵⁵.

In plant metabolism, glucose esters are commonly used as acyl donors. This involves a glucose moiety being added to a carboxylic acid, and the glucose ester product undergoes transesterification, a step catalysed by serine carboxypeptidase-like acyltransferases (SCPL-ATs)⁵⁶. This is analogous to coenzyme A thioesters, which are acyl donors that undergo transesterification catalysed by BAHD-acyltransferases⁵⁶. Glucose ester intermediates feature in the biosynthesis of sinapate (sinapoylcholine) and the UV-protectant sinapoylmalate from sinapic acid via the activated sinapoylglucose ester intermediate in Brassicaceae⁵⁷, as well as in hormone homeostasis forming IAA-myo-inositol conjugates via indoleacetic acid-glucoside⁵⁸. Interestingly, a glucoside ester is an intermediate in tropane alkaloid biosynthesis, a pathway from Solanaceae species that shares many features with nicotine biosynthesis. In particular, a UGT glucosylates phenyllactate to phenyllactylglucoside, enabling SCPL-AT catalysed transesterification to littorine with tropine⁵⁹. The glucosyl group in glucose esters acts to activate the carboxylic acid by forming an ester and becoming a favourable leaving group, with glucose being a product of the transesterification reaction. This is distinct from the proposed glucose activation in nicotine biosynthesis where the glucose is not a direct product of the core reaction but instead modulates the activity of the intermediates and requires a dedicated removal step: NicGH catalysed hydrolysis.

A cryptic glucosylation was recently described in Solanaceae steroidal glycoalkaloid biosynthesis, involving the cholesterol glucuronosyl transferase GAME15^{60,61}. A cryptic glucosylation was also discovered in the biosynthesis of the yeast antibiotic anisomycin⁶². In both

of these examples it appears that the glycosyl moiety acts mostly as a protecting group/recognition handle for the downstream biosynthetic enzymes, rather than having an activation effect as we propose for nicotine biosynthesis.

As well as modulating reactivity, the detoxification effect of glucoside-conjugates in plants can be attributed to driving vacuolar localisation of metabolites. This may also be true for nicotine biosynthesis, given the known subcellular localisation of NaGR (A622) to the cytosol³², and NicGS (BBL) to the vacuole³⁰, and also given that nicotine accumulates in vacuoles of tobacco root cells during biosynthesis³⁸. The predicted *N*-terminal signal peptide and *C*-terminal vacuolar sorting signal of NicGH strongly suggest it will also localise to the vacuole (Fig. S1). The recent manuscript reporting tobacco β -glucosidases included an experiment where GFP was fused to the *C*-terminus of β -GD2 (NtBGLU50) and the fusion protein was transiently expressed in *N. benthamiana*. Surprisingly, the protein appeared to localise to the endoplasmic reticulum, despite its predicted vacuolar localisation sequence. Further investigation of subcellular localisation of β -GDs is required for a conclusive outcome.

Overall, this is an excellent and impactful study that significantly advances our understanding of nicotine biosynthesis. With minor clarifications and expanded discussion, it will be more accessible and compelling to the community.

Thank you!

References

- Brent Friesen, J., and Edward Leete. 1990. "Nicotine Synthase - an Enzyme from *Nicotiana* Species Which Catalyzes the Formation of (*S*)-Nicotine from Nicotinic Acid and 1-Methyl- δ 'pyrrolinium Chloride." *Tetrahedron Letters* 31 (44): 6295–98.
- Kajikawa, Masataka, Nobuhiro Hirai, and Takashi Hashimoto. 2009. "A PIP-Family Protein Is Required for Biosynthesis of Tobacco Alkaloids." *Plant Molecular Biology* 69 (3): 287–98.
- Kajikawa, Masataka, Tsubasa Shoji, Akira Kato, and Takashi Hashimoto. 2011. "Vacuole-Localized Berberine Bridge Enzyme-like Proteins Are Required for a Late Step of Nicotine Biosynthesis in Tobacco." *Plant Physiology* 155 (4): 2010–22.
- Liu, Junhong, Huabing Liu, Zefeng Li, Junping Gao, Peijian Cao, Jun Yang, and Xiaodong Xie. 2026. "Systematic Genomic Characterization of β -Glucosidase Genes in Tobacco Reveals NtBGLU50 as a Multifunctional Regulator of Glycoside Metabolism and Defense-Associated Pathways." *Industrial Crops and Products* 240 (122643): 122643.
- Shoji, Tsubasa, Robert Winz, Tadayuki Iwase, Keiji Nakajima, Yasuyuki Yamada, and Takashi Hashimoto. 2002. "Expression Patterns of Two Tobacco Isoflavone Reductase-like Genes and Their Possible Roles in Secondary Metabolism in Tobacco." *Plant Molecular Biology* 50 (3): 427–40.